# Insights into the ecological roles and evolution of methyl-coenzyme M reductase-containing hot spring Archaea

Zheng-Shuang Hua[1,2,14], Yu-Lin Wang[3,14], Paul N. Evans[4,14], Yan-Ni Qu[1], Kian Mau Goh [5], Yang-Zhi Rao [1], Yan-Ling Qi[1], Yu-Xian Li[1], Min-Jun Huang[2], Jian-Yu Jiao[1], Ya-Ting Chen[1], Yan-Ping Mao[3,6], Wen-Sheng Shu[7], Wael Hozzein [8,9], Brian P. Hedlund [10,11], Gene W. Tyson [4,12]*, Tong Zhang[3]* & Wen-Jun Li [1,13]*

Several recent studies have shown the presence of genes for the key enzyme associated with archaeal methane/alkane metabolism, methyl-coenzyme M reductase (Mcr), in metagenome-assembled genomes (MAGs) divergent to existing archaeal lineages. Here, we study the *mcr*-containing archaeal MAGs from several hot springs, which reveal further expansion in the diversity of archaeal organisms performing methane/alkane metabolism. Significantly, an MAG basal to organisms from the phylum *Thaumarchaeota* that contains *mcr* genes, but not those for ammonia oxidation or aerobic metabolism, is identified. Together, our phylogenetic analyses and ancestral state reconstructions suggest a mostly vertical evolution of *mcrABG* genes among methanogens and methanotrophs, along with frequent horizontal gene transfer of *mcr* genes between alkanotrophs. Analysis of all *mcr*-containing archaeal MAGs/genomes suggests a hydrothermal origin for these microorganisms based on optimal growth temperature predictions. These results also suggest methane/alkane oxidation or methanogenesis at high temperature likely existed in a common archaeal ancestor.

---

[1] State Key Laboratory of Biocontrol, Guangdong Provincial Key Laboratory of Plant Resources and Southern Marine Science and Engineering Guangdong Laboratory (Zhuhai), School of Life Sciences, Sun Yat-Sen University, 510275 Guangzhou, PR China. [2] Department of Biological Sciences, Dartmouth College, Hanover, NH 03755, USA. [3] Environmental Microbiome Engineering and Biotechnology Laboratory, Department of Civil Engineering, The University of Hong Kong, 999077 Hong Kong, SAR, PR China. [4] The Australian Centre for Ecogenomics, School of Chemistry and Molecular Biosciences, University of Queensland, St Lucia 4072 QLD, Australia. [5] Faculty of Biosciences and Medical Engineering, Universiti Teknologi Malaysia, Skudai, Johor 81310, Malaysia. [6] College of Chemistry and Environmental Engineering, Shenzhen University, 518060 Shenzhen, PR China. [7] School of Life Sciences, South China Normal University, 510631 Guangzhou, PR China. [8] Bioproducts Research Chair, Zoology Department, College of Science, King Saud University, Riyadh 11451, Saudi Arabia. [9] Botany and Microbiology Department, Faculty of Science, Beni-Suef University, Beni-Suef 65211, Egypt. [10] School of Life Sciences, University of Nevada Las Vegas, Las Vegas, NV 89154, USA. [11] Nevada Institute of Personalized Medicine, University of Nevada Las Vegas, Las Vegas, NV 89154, USA. [12] Advanced Water Management Centre, University of Queensland, St Lucia 4072 QLD, Australia. [13] Key Laboratory of Biogeography and Bioresource in Arid Land, Xinjiang Institute of Ecology and Geography, Chinese Academy of Sciences, 830011 Urumqi, PR China. [14]These authors contributed equally: Zheng-Shuang Hua, Yu-Lin Wang, Paul N. Evans. *email: g.tyson@uq.edu.au; zhangt@hku.hk; liwenjun3@mail.sysu.edu.cn

Methane is an important compound in both the abiotic and biotic cycling of carbon on Earth. While the majority of biological methane formation is mediated by certain archaea[1], it has also been suggested that aerobic bacteria are responsible for a limited amount of methane biosynthesis[2]. Likewise, methane can be oxidized by certain lineages of Archaea in anaerobic environments[3], with bacterial methanotrophs also responsible for methane oxidation in aerobic and some anaerobic environments[4,5]. The archaea that mediate methane metabolism in anaerobic environments are dominated by organisms from a small number of lineages within the phylum *Euryarchaeota* that share a core metabolism centered around the methyl-coenzyme M reductase complex (Mcr)[6]. This complex catalyzes the terminal step in methanogenesis and initial step in methane oxidation[1]. While these *mcr*-containing organisms that have been shown to produce methane are restricted to the *Euryarchaeota*, there is much debate on the evolutionary origin of this complex[7]. More recently, genes encoding for the predicted metabolism of methane, including divergent *mcr* genes, were identified in two metagenome-assembled genomes (MAGs) from the archaeal phylum *Bathyarchaeota*[8]. Further studies have suggested that these divergent *mcr* genes oxidize alkanes other than methane based on the detection of metabolic intermediates predicted to be formed from propane and butane oxidation by the Mcr complex[9]. Subsequently, other archaeal MAGs from non-euryarchaeal lineages have been discovered that were found to contain *mcr* genes similar to cultivated methanogens[10]. These *Verstraetearchaeota* MAGs have been previously observed in a cultivation-independent survey of *mcrA* genes in several environments[11], before they were taxonomically linked to MAGs[10]. Recently other MAGs that harbor *mcr* genes have been identified from archaeal lineages not previously associated with methane or alkane metabolisms[12–17]. Together these results further consolidate the wide distribution of *mcr*-mediated methane metabolizing archaea beyond our current knowledge, suggesting that the *mcr* catalysis should be expanded to include alkane oxidation[9,18], and underscoring the need for the axenic cultivation of these organisms[8].

Here, we successfully reconstruct MAGs containing *mcr* gene clusters using bioinformatic methods from metagenomes of hot spring microbial communities, which represent potential methanogens, methanotrophs, or alkanotrophs. Phylogenomic analysis shows that these genomes branch within the TACK superphylum and *Euryarchaeota* phylum, but often form distinct clades divergent from cultivated lineages. Similarly, metabolic reconstructions of these organisms inferred from high-quality MAGs suggest unreported methane and alkane metabolism pathways not seen in previous genomes. Also, evolutionary analyses of these and database MAGs revealed methanogenesis or methane oxidation may be a primordial form of energy metabolism in early free-living archaea. The ancestral state of optimal growth temperature indicated these lineages represent an ancient metabolic capacity that originated from thermal habitats. Overall, this study significantly expands our current knowledge about the diversity of methanogens, methanotrophs and alkanotrophs, summarizes recently identified lineages, and sheds further light on the evolution of Mcr-mediated alkane metabolism within the Archaea.

## Results and discussion

**Discovery of *mcr*-containing archaeal MAGs in hot springs.** A total of 14 *mcr*-containing archaeal MAGs were successfully reconstructed from six metagenomes that were sequenced from hot spring sediments located in Tengchong County of Yunnan, China (Table 1). Nine *mcr*-containing MAGs were assembled

**Table 1 Summary statistics of the methanogenic and methanotrophic archaeal bins reconstructed from hot spring samples**

| Bins | Nezhaarchaeota | | | | Verstraetearchaeota | | | | | Euryarchaeota | | | | Thaumarchaeota |
|---|---|---|---|---|---|---|---|---|---|---|---|---|---|---|
| | JZ bin_38 | JZ-1 bin_66 | GMQP bin_37 | ZMQR bin_18 | JZ-2 bin_200 | JZ-3 bin_106 | JZ-3 bin_107 | GMQP bin_44 | DRTY-6 bin_144 | JZ-1 bin_103 | JZ-2 bin_168 | JZ-2 bin_199 | GMQP bin_32 | JZ-2 bin_220 |
| No. of scaffolds | 11 | 3 | 2 | 6 | 38 | 49 | 48 | 7 | 67 | 63 | 10 | 20 | 20 | 54 |
| Genome size (Mbp) | 1.51 | 1.52 | 1.49 | 1.27 | 1.25 | 1.33 | 1.06 | 1.17 | 1.20 | 0.95 | 1.61 | 1.24 | 1.73 | 1.23 |
| GC content (%) | 43.7 | 43.6 | 45.1 | 45.2 | 47.0 | 46.7 | 46.0 | 28.1 | 46.0 | 55.8 | 43.7 | 62.5 | 41.3 | 39.0 |
| No. of protein coding genes | 1571 | 1574 | 1595 | 1354 | 1390 | 1493 | 1224 | 1306 | 1369 | 1060 | 1563 | 1336 | 1967 | 1395 |
| Coding density (%) | 88.5 | 88.4 | 88.2 | 87.9 | 90.1 | 89.8 | 90.1 | 93.3 | 88.4 | 95.0 | 86.7 | 94.8 | 92.8 | 91.9 |
| No. of rRNAs | 2 | 3 | 3 | 3 | 1 | 3 | 1 | 3 | 3 | 1 | 3 | 1 | 2 | 0 |
| No. of tRNAs | 42 | 44 | 43 | 44 | 32 | 39 | 29 | 43 | 39 | 35 | 36 | 38 | 42 | 31 |
| No. of genes | 1146 | 1140 | 1121 | 939 | 935 | 989 | 794 | 948 | 850 | 731 | 1113 | 941 | 1376 | 927 |
| No. of genes annotated by COG[a] | 873 | 871 | 876 | 736 | 727 | 752 | 622 | 755 | 648 | 564 | 871 | 727 | 1063 | 723 |
| No. of genes annotated by KO[a] | 871 | | | | | | | | | | | | | |
| Completeness (%)[b] | 97.55 | 99.02 | 98.53 | 98.53 | 92.06 | 90.65 | 81.31 | 100 | 82.24 | 81.15 | 99.6 | 93.46 | 100 | 97.09 |
| Contamination (%)[b] | 2.21 | 1.47 | 2.21 | 0.74 | 0.06 | 0.07 | 0.05 | 0 | 2.8 | 0 | 0 | 0 | 0.98 | 0.97 |
| Relative abundance (%)[c] | 0.50 | 0.47 | 0.75 | 0.46 | 0.06 | 0.07 | 0.05 | 0.08 | 0.47 | 0.29 | 0.08 | 0.07 | 0.19 | 0.07 |

[a]Functional annotations for all genomes were conducted by uploading to IMG database
[b]Genome completeness and contamination were estimated using CheckM[61]
[c]The relative abundance of each bin was calculated as: total reads mapped to bins/total reads in corresponding sample

from JinZe (JZ) hot springs, which were collected from three different sites (named JZ-1, -2, -3). An additional three *mcr*-containing MAGs from GuMingQuan (GMQ), one from DiReTiYanQu-6 (DRTY-6) and one from ZiMeiQuan (ZMQ) hot spring sediments were also reconstructed. These hot springs are high-temperature (60–98 °C) environments that range from near-neutral to alkaline pH values (6–9.6) (Supplementary Table 1). Most MAGs were classified as high quality with sizes ranging from 0.95 to 1.73 Mbp, with completeness estimates of greater than 90% (Supplementary Fig. 1; Table 1). Both rRNAs and tRNAs (>29) are detectable in nearly all MAGs and is consistent with these MAGs being of high quality[19]. The key taxonomic marker genes for archaeal methane and alkane metabolism, methyl-coenzyme reductase (*mcrABG*), were identified in each of these 14 MAGs on scaffolds with even sequence depth at medium to high coverage (Supplementary Fig. 2). In most cases, each *mcr* complex is located on a long scaffold (>50 Kbp) with several nearby genes annotated as being associated with methane metabolism, as well as tRNAs, chaperon proteins and/or ribosomal proteins interspersed within the corresponding scaffolds (Supplementary Fig. 3). In combination, these results strongly indicate the high accuracy of assembled *mcr*-containing scaffolds and the identified *mcr* genes do belong to these reconstructed MAGs. Phylogenomic analysis showed that these MAGs are widely distributed across both the TACK superphylum (10 MAGs) and the *Euryarchaeota* phylum (four MAGs) (Fig. 1; Supplementary Data 1). Taxonomic placement revealed that four of the 10 TACK-associated MAGs form a clade that is sister to the

*Nezhaarchaeota* representing a new order within this phylum (JZ bin_38, JZ-1 bin_66, GMQP bin_37 and ZMQR bin_18); five belong to *Verstraetearchaeota* (JZ-2 bin_200, JZ-3 bin_106, JZ-3 bin_107, GMQP bin_44 and DRTY-6 bin_144); and the last one branches deeply within the *Thaumarchaeota* (JZ-2 bin_220). The four euryarchaeal MAGs were found to belong to the *Hadesarchaeota* (JZ-1 bin_103 and JZ-2 bin_199), *Methanomassilii-coccales* (JZ-2 bin_168), and *Archaeoglobales* (GMQP bin_32) lineages (Fig. 1). The placement of these and other *mcr*-containing MAGs reveals deep branching within the respective lineages, and in the case of *Thaumarchaeota*, *Nezhaarchaeota*, *Hadesarchaeota*, *Verstraetearchaeota*, and *Archaeoglobi* MAGs, they are basal to the last common ancestor of their respective phyla (Fig. 1). This result also indicates that the metabolic features of these organisms may predate the metabolism of the more derived, well studied methane metabolizing organisms from the *Euryarchaeota*.

**Methane or alkane metabolism in hot spring Archaea.** The five *mcr*-containing MAGs identified as *Verstraetearchaeota* were each approximately ~1.5 Mb and are similar in size to previously described *Verstraetearchaeota* MAGs[10]. These MAGs are also predicted to be $H_2$-dependent methylotrophic methanogens based on the absence of a complete archaeal Wood-Ljungdahl carbon dioxide reduction pathway and the presence of methyltransferase/coronoid proteins (Fig. 2). Furthermore, the *mcr* genes derived from these hot spring organisms form a clade with *mcr* genes from other characterized $H_2$-dependent methanogens from the clades

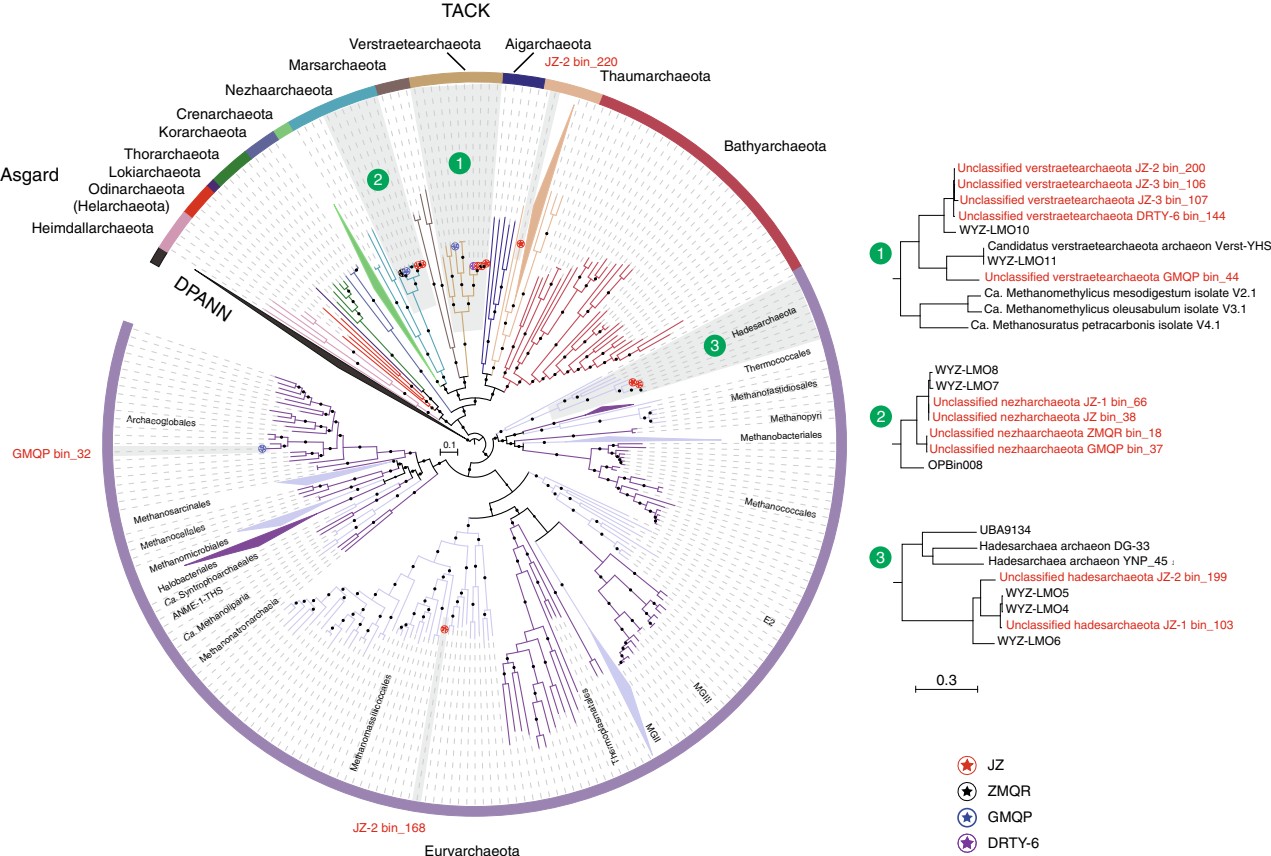

**Fig. 1** The phylogeny of reconstructed methanogenic and methanotrophic MAGs. Maximum-likelihood tree based on 892 archaeal genomes including 14 MAGs in this study was inferred from a concatenated set of 122 proteins using IQ-TREE[69] with 1000 ultrafast bootstrapping iterations. Support values >70% are shown as black circles. Stars in circles represent the MAGs reconstructed in this study. The MAG OPBin_054 of Berghuis et al.[12] that was suggested to belong to the clade with JZ bin_38, JZ-1 bin_66, GMQP bin_37, and ZMQR bin_18 has been excluded from this analysis because it is only 17% complete and is taxonomically difficult to place

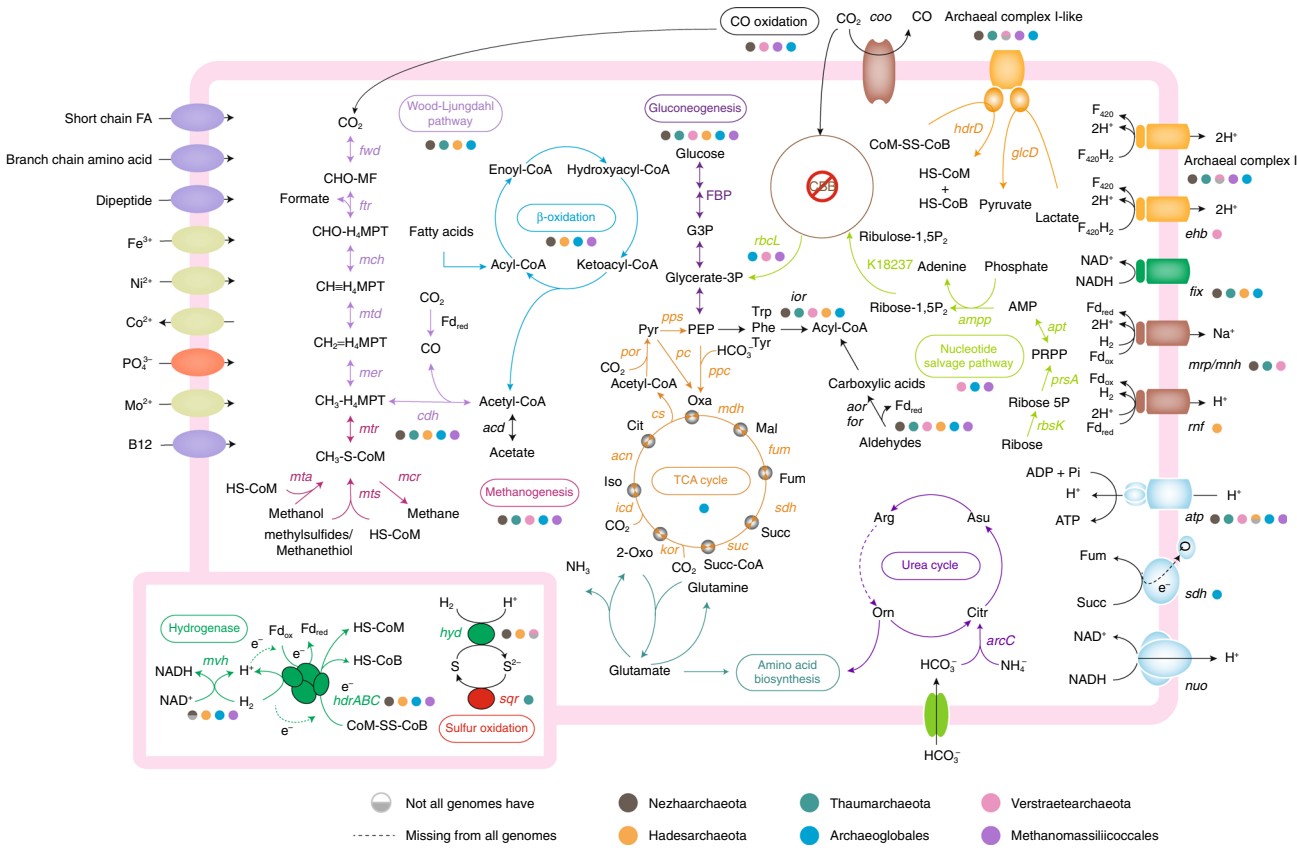

**Fig. 2** Overview of metabolic potentials in *mcr*-containing MAGs. The detected pathways and genes related to metabolisms of carbon, hydrogen, sulfur, and other metabolic pathways including sugar and amino acid utilization, energy conservation, and various transporters in MAG groupings. Different colors indicate separate metabolic modules. Detailed gene copy information associated with above mentioned pathways was recorded in Supplementary Data 3. roTCA reversed oxidative tricarboxylic cycle, CBB Calvin–Benson–Bassham cycle, Fd ferredoxin, PEP phosphoenolpyruvate, G3P glyceraldehyde-3P, PRPP 5-Phospho-alpha-D-ribose 1-diphosphate, FBP fructose-1,6-bisphosphatase

*Methanomassiliicoccales*, *Methanonatronarchaeia*, *Methanofasti-diosales*, and existing *Verstraetearchaeota* genomes (Fig. 3a; Supplementary Data 2). Based on the presence of genes for methyltransferases and corrinoid proteins, the hot spring *Verstraetearchaeota* may form methane from methylated amines, methanol, methylsulfides, methanethiol, or other unknown methylated compounds (Fig. 2). Although, the absence of pyrrolysine biosynthesis and methylamine methyltransferase (*mtbA*) genes suggest that methane formation from methylated amines is unlikely to occur. It is likely that methanogenesis from methylsulfides, methanethiol, or methanol occurs based on the presence of *mtsA* and *mtaA* genes in these *Verstraetearchaeota* MAGs (Fig. 2). Methylsulfides and methanethiol are common sulfur species observed in thermal springs[20]. Congruent with other $H_2$-dependent methylotrophic methanogens is the absence of genes for a carbon dioxide reduction pathway and tetrahydromethanopterin S-methyltransferase (*mtrABCDEFGH*) that would allow hydrogenotrophic methanogenesis to proceed[21]. While the complete *mtr* operon is absent, the presence of predicted corrinoid containing *mtrH* subunit homologs with adjacent methyltransferases may allow this organism to form methane from unknown methylated compounds. The absence of heterodisulfide reductase (*hdrABC*) subunits in *Verstraetearchaeota* MAGs (Supplementary Data 3) suggests that mechanisms for the regeneration of coenzyme M and B may be carried out by a proposed *hdrD-glcD* complex (Fig. 2). However, the presence of *fpo* genes (archaeal complex I) suggests energy is likely conserved

in a similar manner to $H_2$-dependent methanogens from the *Methanomassiliicoccales*, *Methanonatronarchaeia*, *Methanofasti-diosales*, and *Verstraetearchaeota*. In these organisms, energy could be conserved by an *hdrD*-like protein that regenerates the coenzyme M and B cofactors, with the concomitant translocation of protons across the membrane[22,23] (Fig. 2). Also, the presence of the key energy-conserving complex *ehb* seen in the coal seam and bioreactor *Verstraetearchaeota* genomes suggests a similar methane-forming metabolism[10]. Core methanogenesis genes (*mcrABGCD*, *hdrD*, *glcD*, *mtsA*, *mtaA*, *mttBC*) are also present in these organisms, and in previously published *Verstraetearchaeota* genomes, suggesting a similar metabolism type between all these organisms. Also, *mcr*-containing $H_2$-dependent methanogens identified in other thermal springs have been suggested to possess a methane-forming metabolism similar to *Methanomassiliicoccales* methanogens[22,24]. The inferred metabolism of these thermophilic *Verstraetearchaeota* organisms is not surprising as methylated compounds[25] and hydrogen[26] are key metabolic intermediates in thermal spring environments.

While $H_2$-dependent methylotrophic *Verstraetearchaeota* and *Methanomassiliicoccales* methanogens appear to be important in these hot spring environments, predicted hydrogenotrophic methanogens that form a clade sister to the *Crenarchaeota* are also present (Fig. 1). These organisms have been observed in several hot springs[12,17] and appear to be the first hydrogenotrophic *mcr*-containing organisms outside the *Euryarchaeota* based on the presence of genes for the archaeal Wood-Ljungdahl pathway,

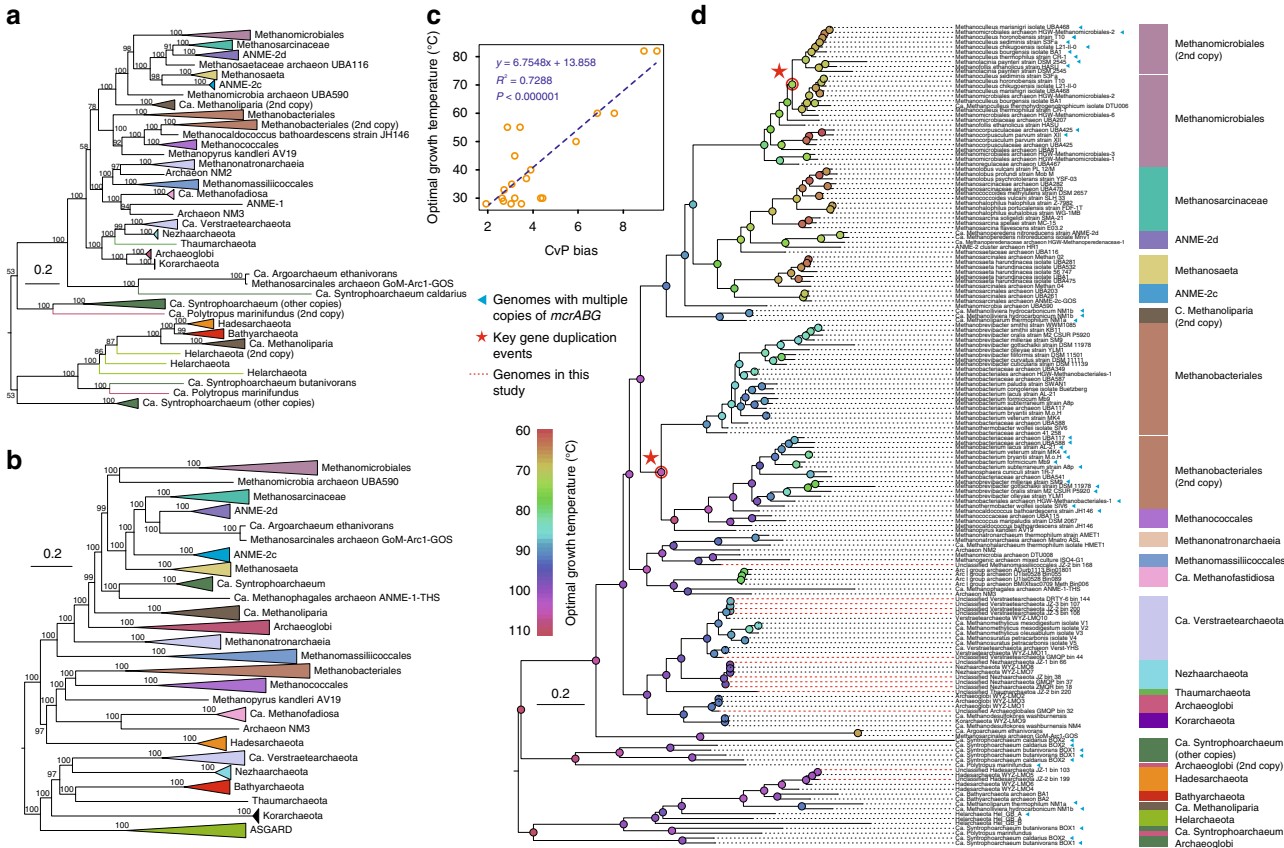

**Fig. 3** Phylogenetic tree of concatenated *mcrABG* genes and optimal growth temperature (OGT) estimation of *mcrABG*-containing microbes. **a** The collapsed maximum-likelihood tree constructed by IQ-TREE represents a phylogeny of methanogens, methanotrophs and alkanotrophs based on an alignment of concatenated McrABG sequences. Bootstrap supports for 1000 iterations are shown on each node. **b** A collapsed phylogenomic tree of *mcr*-containing microbes which were selected from the maximum-likelihood tree in Fig. 1. **c** Linear regression analysis was conducted to model the relationship between charged versus polar amino acid ratios (CvP bias) and current known OGTs of methanogens (See Methods for detailed description). $R^2$ value is equal to 0.7288 with a *p*-value of <0.000001. **d** The expanded phylogeny of the McrABG has the similar topological structure as **a** without listing the bootstrap values on each node. Branch lengths are re-estimated during the ancestral sequence reconstruction in PAML[45]. Color keys on the nodes indicate the estimated OGTs. Dashed lines in red show the MAGs/genomes presented in this study. Red stars show potential ancient gene duplication events. Blue triangles indicate McrABG sequences that are present in a MAG/genome in multiple copies

methyl-coenzyme M reductase complex (*mcr*; *mcrABGCD*), all subunits of tetrahydromethanopterin S-methyltransferase (*mtr*; *mtrABCDEFGH*), and heterodisulfide reductase (*hdr*; *hdrABC*) (Supplementary Data 3). We also show there is a greater diversity of these organisms from the same samples compared to Wang et al.[17] (Fig. 1); however, there is little difference in the metabolic capabilities of these organisms based on gene complements (Fig. 2; Supplementary Data 3).

Another *mcr*-containing MAG, GMQP bin_32, with a similar metabolic potential to the *Nezhaarchaeota*-like MAGs presented here and in Wang et al.[17] was found to belong to the order *Archaeoglobales* (Fig. 1). However, the MAG only shared ~60% amino acid identity to the closest subgroup of organisms from the genus *Archaeoglobus* (Supplementary Fig. 4a). The similarity of the GMQP bin_32 MAG to the *Nezhaarchaeota*-like MAGs suggests it is also a hydrogenotrophic methanogen consistent with the results of Wang et al.[17]. This would also be consistent with speculation that most *Archaeoglobales* lost the metabolic capability for methane metabolism[27], but is intriguing as recent analysis of another *Archaeoglobales* genome (JdFR-42) that harbored *mcr* genes was predicted to perform alkane oxidation[14,17]. Therefore, the ability for methanotrophy in GMQP bin_32 and the *Nezhaarchaeota*-like MAGs cannot be ruled out (Fig. 2). Together these results also suggest that the last common

ancestor of these *Archaeoglobales* organisms possibly possessed the Mcr complex.

A further MAG from outside the *Euryarchaeota* was found to cluster with the *Thaumarchaeota* and had a combination of methane metabolism genes that were distinct from other *mcr*-containing hot spring organisms (Fig. 2). Based on the presence of genes for methyltransferases and corrinoid proteins (*mttBC*, *mtaA*, and *mtsA*), along with the absence of a complete *mtrABCDEFGH* operon, it is suggested that this organism is a $H_2$-dependent methylotrophic methanogen. Consistent with its predicted methylotrophic lifestyle, the *Thaumarchaeota* organism does possess *mtrAH* genes suggesting metabolism of unknown methylated compounds could occur. Furthermore, genes that encode for an incomplete archaeal complex I suggests an electron carrier other than $F_{420}$, such as ferredoxin seen in *Methanothrix* organisms[28].

While many of these hot spring microbial community MAGs appear to have conventional *mcr* genes associated with methanogenesis or methanotrophy (Fig. 2). A group of archaea associated with *Hadesarchaeota* MAGs contain *mcr*-like genes similar to *Bathyarchaeota* and *Ca*. Syntrophoarchaeum organisms that have been associated with Mcr-mediated alkane oxidation[9,29] (Fig. 3a). These *Hadesarchaeota* MAGs also possess genes associated with the archaeal Wood-Ljungdahl, β-oxidation, and

other pathways similar to those found in other *mcr*-containing *Hadesarchaeota* genomes that have been recently identified[17]. Previously identified non-*mcr*-containing *Hadesarchaeota* genomes were reported to have the ability to metabolize sugars and amino acids in a heterotrophic lifestyle, oxidize carbon monoxide (*coxMLSIF*), and perform dissimilatory nitrite reduction to ammonium[30,31]. These current *mcr*-containing *Hadesarchaeota* MAGs in this study do not contain these pathways. While Wang et al.[17] identified only a single copy of a *mcrA* gene in each of their *Hadesarchaeota* MAGs, a second copy of *mcrA* in a single *mcr* operon was identified in the analysis of our MAGs (Supplementary Figs. 2 and 3). This operon structure may be the result of gene duplication and may lead to the neofunctionalisation of the Mcr complex to metabolize short chain alkanes of different lengths suggested recently by Evans et al.[32].

The expansion of *mcr* gene diversity from the MAGs presented here and in other studies[12–15,17] reveals organisms with relatively similar *mcr* complexes from across the archaeal species tree (Fig. 3a, d). These organisms have characteristics of either H$_2$-dependent methylotrophic or hydrogenotrophic methanogens, although given the similarity to pathways from Anaerobic Methanotrophs (ANME), the ability of these organisms to oxidize methane cannot be ruled out. Overall, the discovery of these many novel MAGs suggests that yet more novel lineages of *mcr*-containing Archaea will be identified in the future, with *mcr*-containing *Asgardarchaeota* being a recent example of this[16].

**Carbon fixation.** Four of the six *mcr*-containing MAG clades (*Nezhaarchaeota*, *Thaumarchaeota*, *Hadesarchaeota*, and *Archaeoglobales*) appear to encode a conventional archaeal Wood-Ljungdahl pathway carbon dioxide fixation pathway, that includes genes for acetyl-CoA synthase/carbon monoxide dehydrogenase (*acs/codh*), and pyruvate oxidoreductase complexes (*porABDG*) (Fig. 2; Supplementary Data 3). This result is consistent with many archaea, including methanogens, which use this pathway to fix carbon[33]. MAGs from *Verstraetearchaeota* and *Methanomassiliicoccales* are thought to be unable to fix carbon and likely use organic carbon, which has been suggested in previous studies[10,34]. Ribulose 1,5- biphosphate carboxylases/oxygenases (RuBisCOs) were identified in *Archaeoglobales*, *Methanomassiliicoccales* and four of the five *Verstraetearchaeota* MAGs (Supplementary Fig. 5). Among them, the *Archaeoglobales* and one of the *Verstraetearchaeota* MAGs harbored form III-b RuBisCOs, suggesting the potential function in a nucleoside-salvaging pathway instead of carbon dioxide fixation via the Calvin–Benson–Bassham (CBB) cycle[35]. The remaining MAGs have RuBisCOs categorized as the form III-a group that is specific to methanogenic archaea. No complete CBB cycle was identified in these MAGs. It was suggested that form III-a RuBisCOs may be involved in the reductive hexulose-phosphate pathway to fix carbon dioxide, with energy and reducing power supported by methanogenic pathways[36]. However, no phosphoribulokinase was detected in any of the MAGs. This may be the result of the high-temperature environment driving genome streamlining, leading to a deficiency in this autotrophic pathway.

Overall, there appears to be a diverse range of mechanisms for fixing carbon in these *mcr*-containing MAGs, reflecting the wide diversity of these lineages across the archaeal tree. This result is congruent with the multiple mechanisms that organisms across the archaea possess ability to fix carbon[37]. Four of the six genome lineages also appear to generate energy from β-oxidation of fatty acids or other linear chain hydrocarbons (Fig. 2). In the case of the *Hadesarchaeota*, the presence of this pathway is not surprising because a modified β-oxidation pathway has been proposed previously in this lineage for Mcr-mediated alkane oxidation[17]. Alkane oxidation has been seen in archaea previously including those belonging to the order *Archaeoglobales*, but is mediated by the addition of fumarate rather than β-oxidation[38].

**Evolution of methane and alkane metabolism.** Previous studies have shown that Mcr-mediated methane and alkane metabolism is widespread across the archaeal tree based on the expanding taxonomy of archaea with these metabolism types[8–10,12–14]. However, it is unclear if this wide diversity of organisms carrying these genes is the result of vertical descent or horizontal gene transfer (HGT), based on the paucity of *mcr* gene-containing lineages within the TACK superphylum[32]. Here, the recovery and analysis of MAGs containing *mcr* genes from hot springs provides stronger evidence that *mcr* within TACK superphylum genomes are the result of vertical descent (Fig. 3a). These lineages include those that are notionally within the *Nezhaarchaeota* (four MAGs), *Verstraetearchaeota* (five MAGs), *Korarchaeota* (three MAGs), and *Thaumarchaeota* (one MAG) (Fig. 1). The *mcr*-containing scaffolds from these MAGs always harbor genes related to methane metabolism including several putative methanogenesis marker proteins, genes related to energy conservation with the exception of *Hadesarchaeota* (Supplementary Fig. 3). In these cases, HGT is unlikely based on the absence of differences in DNA sequence composition as there is little variation in the *mcr*-containing sequence composition above the natural variation in their respective MAGs/genomes (Supplementary Table 2). For the *mcr*-containing TACK lineages that fall within or sister to *Verstraetearchaeota* MAGs, the genome and *mcr* gene taxonomies appear to be congruent (Figs. 1 and 3a) and strengthens the case for the *mcr* complex being present in the common ancestor of the TACK and *Euryarchaeota* lineages[32]. The presence of the euryarchaeal *Archaeoglobus* GMQP bin_32 MAG Mcr sequences sister to the *Nezhaarchaeota* and H$_2$-dependent methanogens (Figs. 1 and 3a) could be explained by HGT. It has been suggested that this mechanism is common in *Archaeoglobi* and has been suggested as a driver of gaining sulfate reduction at the expense of methane or alkane metabolism in these organisms[14,17,32,39] (Supplementary Fig. 4b). Based on the paucity of *Archaeoglobi* MAGs or genomes containing *mcr* genes, the taxonomic resolution of these analyses is limited and requires further representatives to understand the complete picture of the evolutionary relationships between these organisms and their *mcr* genes.

In the case of *mcr* genes in the *Bathyarchaeota*, *Archaeoglobi*, *Ca.* Argoarchaeum, *Helarchaeota*, *Hadesarchaeota*, NM1, and *Ca.* Syntrophoarchaeum MAGs, it is likely that these organisms perform alkane oxidation[9,13,16,18]. With the exception of the *Helarchaeota* and *Bathyarchaeota*, these *mcr* genes are exclusively found in euryarchaeal lineages (Fig. 1). Therefore, it is possible that HGT of the *mcr* genes from a euryarchaeal ancestor to the *Helarchaeota* and *Bathyarchaeota* is the reason for its presence in the Asgardarchaeota and TACK lineages, respectively[16,32]. This also suggests the origin of the *mcr* genes associated with alkane oxidation is likely to be within the *Euryarchaeota* and the long branches seen in this clade are likely the result of neofunctionalisation arising from gene duplication (Fig. 3a). HGT may also play a role in the propagation of Mcr-mediated alkane oxidation within the *Euryarchaeota*, based on the observation of high variation of GC content seen in the *mcr*-containing scaffold of *Hadesarchaeota* JZ-1 bin_103 compared to other scaffolds of this MAG (Supplementary Table 2). While HGT as a mechanism in Mcr evolution remains speculative, the different branching order observed between McrABG sequences (Fig. 3a) and species tree marker proteins (Fig. 3b) for *Ca.* Argoarchaeum and GoM-ArcI-GOS organisms suggest HGT may have occurred. Similarly, the

branching order differences for the methane metabolizing *Ca.* Methanophagales ANME-1-THS and *Ca.* Methanofastidosa lineages also suggest a potential HGT event (Fig. 3a, b).

Further expansion of *mcr*-containing organisms to include the *Thaumarchaeota* associated JZ-2 bin_220 MAG suggests an interesting evolutionary story in the context of this metabolism type. Phylogenomic analysis places this MAG close to the pSL12 lineage (DRTY-7 bin_36)[40] that does not contain *mcr* genes (Supplementary Fig. 6a). Unlike other *Thaumarchaeota* genomes and previously cultured isolates, JZ-2 bin_220 does not contain genes for ammonia oxidation (*amoABC*) or aerobic respiration (aa₃, bd, cbb3, and bo₃-cytochromes) and suggests that the ancestor of *Thaumarchaeota* was an anaerobe with the ability for methane metabolism. Along with ancestral gene changes, JZ-2 bin-220 shows a significant alteration in gene complement with the acquisition of 457 and loss of 194, genes that likely led to the divergence of these aerobic and anaerobic lineages (Supplementary Fig. 6b; Supplementary Data 4). Consistent with the metabolism of the *mcr*-containing *Thaumarchaeota* MAG being the result of vertical descent, is the basal nature of this genome within the *Thaumarchaeota* and the presence of many genes associated with methane metabolism (Supplementary Fig. 6c; Supplementary Data 4). Many of these genes, excluding *mcr*, are also found in the *Aigarchaeota* MAGs RBG_16_49_8, JGI MDM2 JNZ-1-N15, and JGI MDM2 JNZ-1-K18. For example, phylogeny of RNA polymerase sequences also places the *Aigarchaeota* into the *Thaumarchaeota* clade with a high bootstrap confidence[40,41]. However, a definitive picture of the evolutionary history of these *Thaumarchaeota* organisms awaits additional genomic data from related organisms.

In the case of the *mcr*-containing *Hadesarchaeota* MAGs, they appear to have a high degree of similarity to *Hadesarchaeota* MAGs without *mcr* genes as 1125 gene families at the nucleotide (Supplementary Fig. 7a) and amino acid (Supplementary Fig. 7b) levels have high homology to each other. The presence of genes indicative of methane metabolism in JZ-1 bin_103 and JZ-2 bin_199 MAGs suggests there has either been a gain of genes by non-*mcr*-containing *Hadesarchaeota* MAGs or gene loss events resulting in the deficiency among non-*mcr*-containing *Hadesarchaeota*. The presence of genes, such as *mtd*, *mch*, along with *hdr* genes suggests an ancestor of extant *Hadesarchaeota* organisms potentially, had the ability for methane or alkane oxidation (Supplementary Fig. 7c; Supplementary Data 4). Alternatively, these genes associated with methane metabolism were already present in the *Hadesarchaeota* MAGs and *mcr* genes and other associated genes were acquired in a manner that is similar to that predicted for *Archaeoglobales* GMQP bin_32 (Supplementary Fig. 4) and *Bathyarchaeota* organisms[29,42].

Together these results reaffirm the results presented in Evans et al.[32] and adds further to the complex interplay between methane- and alkane-metabolizing archaea due to vertical descent, gene duplication and neofunctionalisation along with some HGT of this metabolism type within the archaea. The now widespread nature of genes encoding the *mcr* complex across the Archaea suggests this metabolism has a central role within archaeal metabolism alongside the similarly widespread and key acetyl-CoA synthase/carbon monoxide dehydrogenase complex[42–44]. This result combined with the presence of archaeal Wood-Ljungdahl carbon dioxide fixation and β-oxidation pathways suggests a shared evolutionary history in these organisms of these metabolism types across the archaeal tree, despite being interspersed with non-*mcr*-containing lineages[32,39].

**Evolution of *mcrABG* genes and their hydrothermal origin.** The similar topologies for *mcrA*, *mcrB*, and *mcrG* genes show that these three genes have similar evolutionary histories (Supplementary Fig. 8a, b, c, d). This result supports the hypothesis that *mcrABG* has coevolved as an integral functional unit[8]. While it is likely *mcrABG* has coevolved, discrepancies in the *mcrG* taxonomy were observed with the methanogenic *Ca.* Verstraetearchaeota is showing higher divergence to most other TACK members instead of existing as a sister lineage to *Nezhaarchaeota* (Supplementary Fig. 8d). This may be an artefact due to the low confidence values at several parent nodes of the TACK lineage and the smaller length of *mcrG* compared to *mcrA* and *mcrB* gene subunits. It seems plausible that Mcr complexes are more conserved in TACK lineages compared to *Euryarchaeota* lineages based on the shorter branch lengths and the presence of only one copy per genome (Fig. 3d). In euryarchaeal methanogen lineages, two copies of the *mcr* operon were observed in *Methanobacteriales* and *Methanomicrobiales* and appear to have evolved in parallel (Fig. 3d; Supplementary Fig. 8). A likely duplication event has also occurred in the *Hadesarchaeota* MAGs presented here based on the organization of a second *mcrA* copy within an existing *mcr* gene operon and the phylogenetic distance between the two copies is small (Supplementary Fig. 8b). Also, duplications of *mcr* genes in the *Ca.* Syntrophoarchaeum and *Archaeoglobi* MAGs appear to be more numerous and ancient[14].

To unwind the evolutionary history of *mcr*-containing microorganisms, we used PAML[45] to computationally reconstruct the ancestral sequences of each internal node. By coupling this with an estimation of the optimal growth temperature (OGT), this gives us an opportunity to infer the environmental temperature of ancestral microbes and better understand their evolutionary history[46]. For the traditional euryarchaeal methanogens, two major ancestral gene duplication events were observed, allowing them to survive in wide range of temperatures. Results show that CvP bias (difference between proportions of charged and polar non-charged amino acids) of concatenated *mcrABG* genes shows significant positive correlation with their OGTs (Person's $R^2 = 0.7288$, $P < 0.000001$; Fig. 3c). By reconstructing the ancestral amino acid sequences for each node, we observed a general pattern that the common ancestor of both methanogens and methanotrophs may have evolved from thermal habitats, as nodes near the root are predicted to be thermophilic. The predicted OGTs for the common ancestors of TACK-belonging methanogens/methanotrophs and alkanotrophs are ~93 and ~100 °C, respectively, which are in line with a thermal origin for archaea[47]. As expected, most extant genomes from hot spring habitats were predicted to be adapted to high temperatures ranging from 63–82 °C, which are comparable to their habitats (Supplementary Table 1). To adapt to the thermal environments, thermophiles evolved both heat stable proteins and increased cell content of heat-shock proteins. Several MAGs contained genes encoding the heat-shock protein *htpX*, as well as high-temperature stress DNA repair systems encoded by *radAB* and RAD51 genes (Supplementary Data 3). All the genomic bins have the predicted capacity to synthesize archaeal membrane lipids. Unlike the bacterial lipid membranes composed of fatty acids linked to glycerol-3-phosphate via ester bonds, archaea possess isoprene-based alkyl chains linked by ether linkages to glyrecerol-1-phosphate, giving cell walls increased stability at high temperature[48]. Also, most bins harbored two copies of reverse DNA gyrase, which could stabilize DNA to cope with the increased temperature[49].

Fossil evidence suggests that geothermal habitats like submarine hydrothermal vents and terrestrial hot springs are a likely place where life on Earth first evolved[50,51], which is also supported by geochemical, biological, thermodynamic and phylogenetic inferences[43,44,52]. On this basis and from the reconstructed archaeal phylogenomic tree (Fig. 1) it is reasonable

to infer that thermophilic methanogenesis or methanotrophy may be a metabolism type associated with early life on earth. Consistent with this result we show the Mcr complex likely has a thermal origin based on ancestral state reconstruction from amino acid residues. Furthermore, it speculated that hydrogenotrophic methanogenesis via the Wood-Ljungdahl pathway might be the most primordial metabolisms for energy conservation and carbon dioxide fixation[52–54]. Those thermophilic *mcr*-containing organisms that branch deeply in the concatenated McrABG tree (Fig. 3a, d) suggest a long evolutionary history of this metabolism type in archaea. This result would be consistent with some of the *mcr*-containing organisms from the TACK lineages harboring pathways for hydrogentrophic methanogenesis could be primordial. Alternatively, by applying phylogenetic analysis of involved key enzymes in acetogenic, methanogenic, and methanotrophic pathways, the same view was proposed for methanotrophy being more likely than Wood-Ljungdahl pathway to date back to the emergence of life[55]. Methane consumed by methanotrophs may come from the abiotic serpentinization, where seawater reacts with olivine at high temperatures to release iron oxide and various carbon compounds[56].

**Summary**. Together these results show that hot spring environments harbor many *mcr*-containing organisms from outside of the *Euryarchaeota* and further expand the diversity of Mcr meditated methane or alkane metabolism processes. The predicted wide range of metabolic mechanisms suggests that these organisms may utilize diverse and as yet unidentified substrates. Here, we propose a plausible evolutionary scenario where the common ancestor of archaea harbors the ability for methane metabolism that is mostly the result of vertical inheritance, with some HGT events. Frequent HGT events have also led to alkanotrophy being found in several lineages that cannot be explained by vertical descent of *mcr* genes. Also, it is also likely that *mcr* gene duplication has led to changes in the substrate specificity to longer chain alkanes in several lineages within the *Euryarchaeota*. We also infer that these *mcr*-containing archaea may originate from thermal habitats such as hydrothermal vents or terrestrial hot springs predicted by a high ancestral optimal growth temperature. Overall, this study enables a better understanding of the origin of last common ancestor in Archaea using a combination of bioinformatic techniques.

## Methods

**Sample collection, DNA extraction, and sequencing**. A total of six biomass samples were collected from four thermal spring sediments (JZ (JinZe), GMQ (GuMingQuan), DRTY (DiReTiYan), and ZMQ (ZiMeiQuan)) located at the collision boundary between the India and Eurasia plates near Tengchong city in Yunnan province (China). These hot springs span a wide range of physiochemical parameters with temperature ranging from 60 to 98 °C and pH ranging from 6.0 to 9.6 (see Supplementary Table 1 for the detailed geographical and physiochemical parameters). Three of the four springs are classified as hydrothermal habitats with temperatures greater than 80 °C. Following collection and DNA extraction, each of the six samples was subjected to metagenomic sequencing. Detailed method for sample collection, DNA extraction, and metagenomic sequencing is described in Hua et al.[40].

**Metagenomic assembly and genome binning**. Metagenomic assembly and genome binning were performed separately for each sample due to the significant differences in microbial community composition among the six samples (Supplementary Fig. 1). Briefly, the raw sequencing data were first preprocessed to remove adapters and duplicated sequences. Low quality reads with average phred value < 20 in a 50 bp sliding window were discarded. Remaining low quality sequences (phred value < 20) were trimmed at both ends. All quality control steps were conducted with custom Perl scripts[57]. The quality reads were assembled using SPAdes[58] (v3.9.0) with the parameters as:–meta -k 21,33,55,77,99,127. GapCloser (v1.12; http://soap.genomics.org.cn/) was used to eliminate the gaps within scaffolds. BBMap (v35.85; http://sourceforge.net/projects/bbmap/) was used to map all the quality reads onto assembled scaffolds with the parameters as: k = 15 minid = 0.9 build = 1. Then, the script "jgi_summarize_bam_contig_depths" in MetaBAT[59]

(v2.12.1) was used to compute the coverage information of each scaffold. Genome binning for each sample was carried on scaffolds using MetaBAT with the coverage information in from each dataset. To further verify the accuracy of the metagenomically assembled genomes (MAGs), emergent self-organizing maps (ESOM)[60] was performed to visualize the bins and scaffolds with abnormal coverage information or discordant positions were removed manually (Supplementary Fig. 1). Subsequently, reads mapped to the curated MAGs were reassembled using SPAdes (v3.9.0) with the following options:–careful -k 21,33,55,77,99,127. Further optimization of each MAGs was conducted as described above. The completeness, contamination and strain heterogeneity of each MAG was evaluated using CheckM[61] (version 1.0.5). A total of 14 MAGs identified as containing *mcrABG* gene sequences from the assembled MAGs were identified using GraftM[62] (v0.10.2).

**Genome annotation and metabolic reconstruction**. Protein coding sequences (CDS) were predicted using Prodigal[63] (v2.6.3) with the "-p single" option. Functional annotation and metabolic reconstructions were performed by querying the predicted CDS against several databases including NCBI-nr, KEGG, eggNOG and CAZy using DIAMOND[64] (v0.7.9; E-values < 1e⁻⁵). Carbohydrate-active enzymes were identified by searching against dbCAN HMMs[65] using HMMer3 (v3.1b2; http://hmmer.janelia.org/). The 14 *mcr*-containing MAGs used in this study were also deposited to Integrated Microbial Genomes (IMG) platform (http://img.jgi.doe.gov) for gene annotation.

**Phylogenetic and phylogenomic analysis**. To understand the phylogenetic relationship of the 14 genomes, a set of 122 conserved marker genes were retrieved from 1213 MAGs/genomes downloaded from public databases which covers nearly the whole archaeal diversity to date to build the phylogenomic tree[41]. Where required, Relative Evolutionary Distances (RED) from the Genome Tree Database project were used for the assignment of taxonomic ranks[41]. Only MAGs of medium to high quality (above a 70% CheckM quality metric) were used in analyses. The marker gene sets were extracted independently using AMPHORA2[66]. All identified marker genes were aligned individually using MUSCLE[67] (v3.8.31). Poorly aligned regions were eliminated by TrimAL[68] (v1.4.rev22; -gt 0.95 -cons 50). Individual alignments were concatenated and used as input to reconstruct the phylogenomic tree using IQ-TREE[69] (v1.6.10) with the mixture model of LG + F + R10 and with ultrafast bootstrapping (-bb 1000), as well as Shimodaira–Hasegawa–like approximate likelihood-ratio test (SH-aLRT, -alrt 1000). The best model determined by ModelFinder[70] is well supported by all criteria including Akaike Information Criterion (AIC), corrected AIC and Bayesian Information Criterion.

For functional genes, including concatenated *mcrABG* genes and individual subunits of these three genes, sequences were aligned using MUSCLE and IQ-tree was used to infer maximum-likelihood phylogenies with the same parameters as above. The best models for *mcrA*, *mcrB*, *mcrG*, and concatenated *mcrABG* genes were LG + F + I + G4, LG + F + R6, LG + I + G4 and LG + F + R6, respectively. We adopted two approaches to root the individual and concatenated *mcrABG* gene trees which place the root at the same place: phylogenetic rooting using the minimal ancestor deviation (MAD) method proposed by Tria et al.[71] and Bayesian tree constructions using BEAST with a lognormal uncorrelated molecular clock[72]. The generated trees in newick format were visualized by iTOL[73] v3.

**Optimal growth temperature (OGT) estimation**. It has been reported that OGT is significantly correlated to charged versus polar amino acid ratios (CvP bias)[74,75]. Charged amino acids include arginine, lysine, aspartic acid, and glutamic acid, while polar amino acids contain glutamine, asparagine, serine and threonine. Here, 304 sequenced methanogens were downloaded from the National Center for Biotechnology Information (NCBI, https://www.ncbi.nlm.nih.gov/) genome repository database. Putative genes were predicted for all genomes using Prodigal[63] with the "-p single -g 11" option. Orthologous genes were identified for each pairwise genomes by extracting the reciprocal best BLAST hits (E-value < 1e-5). Average amino acid identity (AAI) was computed as the mean similarity of all orthologous genes. Single representatives were manually selected for subsequent analysis when two or more genomes showed an average AAI > 97%. A total of 117 genomes were used for OGT analyses including 25 from newly reported MAGs in recently published papers[12–18]. Taxonomic information, sampling site and OGT of genomes were manually investigated from original publications and from the website of the German Collection of Microorganisms and Cell Cultures (http://www.dsmz.de/), if available (Supplementary Data 5). Two psychrophiles were excluded since they evolved distinctly[74]. The distribution of OGTs among these methanogens shows that 18 species out of 37 have an OGT of 37 °C. To overcome this bias, the mean value of those CvPs was calculated for species with OGT of 37 °C. Linear regression was conducted based on the known OGTs and CvP biases of *mcrABG* genes. Species with two copies of *mcr* complex were computed twice with the same OGT. Significant correlation was observed between them by fitting the formula: $y = 6.7548 \times + 13.858$, with Pearman's $R^2 = 0.7288$ and $P < 0.000001$ (Fig. 3c). Key genes including *mcrA*, *mcrB* and *mcrG* were searched against in-house database using AMPHORA2.

**Ancestral amino acid sequence reconstruction**. Amino acid sequences at all internal nodes were computationally inferred based on the sequence alignment and tree topology of concatenated *mcrABG* gene (as described above) using *codeml* program in PAML[45] package (v4.9 h). An empirical Bayesian statistical framework incorporated Gamma distribution was employed for the inference of posterior amino acid probability per site and no molecular clock was set. The universal code (icode = 0) and fixed branch length option (Mgene = 0), as well as 10 categories in the ω distribution under LG model (suggested by ModelFinder as described above) were used throughout the PAML calculation. Both marginal and joint posterior probabilities were calculated in this program.

**Comparative genomics**. Comparative genomic analyses for *Hadesarchaeota*, *Thaumarchaeota*, and *Archaeoglobales* were conducted separately. Reference genomes for each group were downloaded from NCBI and IMG-M (https://img.jgi. doe.gov/cgi-bin/m/main.cgi) databases (Supplementary Data 6). Genome quality was estimated by CheckM and only genomes with completeness >80% and contamination <10% were taken into consideration for the later analysis. Due to the lack of representatives, all MAGs from *Hadesarchaeota* were included for the later analysis even their completeness <80%. Average amino acid identity of each pair of genomes and clusters of homologous proteins were computed as described previously[40]. Briefly, all-against-all genomes BLAST for all the genomes in each group were performed to yield reciprocal best BLAST hits (rBBHs). MCL[76] algorithm (v14–137) was subsequently employed to cluster rBBHs into protein clusters. Bayesian trees were constructed using MrBayes[77] (v3.2.6) by running four independent chains and two simultaneous runs in order to calculate the convergence diagnostics. These analysis starts with 0.1, 0.5, and 1 million generations for lineages of *Hadesarchaeota*, *Archaeoglobi* and *Thaumarchaeota* and the same burn-in fraction of 0.25, as well as sampling every 100 generations and computing diagnostics every 1000 generations for each. Theoretically, the average standard deviations of split frequencies <0.001, or approaching to 0 for the Markov chain Monte Carlo sampling were treated as convergent. Then evolutionary histories for each group were inferred using COUNT[78] (v9.1106) as described previously[40]. Specifically, the rate models were calculated and optimized under the gain–loss–duplication model with the Poisson distribution at the root. The rate of variation across families was set to 1:1:1:1 gamma categories for the edge length, the loss rate, gain rate, and the duplication rate, respectively. The convergence criteria applied were set to 100 rounds for the optimization rounds with a likelihood threshold of 0.1. The family histories for different groups were determined under the Dollo parsimony assumption.

**Nomenclature of MAGs in this study**. Based on the data presented above, we propose the following taxonomic epithets "*Ca*. Methanohydrogenotrophilum pristinum" (JZ bin_68, JZ bin_38), "*Ca*. Geoarchaeum hydrogenovorans" (ZMQR bin_18, GMQP bin_37), "*Ca*. Methanomethylovorus thermophilus" (JZ-2 bin_200, JZ-3 bin_106, JZ-3 bin_107, DRTY-6 bin_144), "*Ca*. Methanomethylarchaeum antiquum" (GMQP bin_44), "*Ca*. Methanourarchaeum thermotelluricum" (JZ-1 bin_103), "*Ca*. Hadesarchaeum tengchongensis" (JZ-2 bin_199), "*Ca*. Methanomixtatrophicum sinensis" (JZ-2 bin_168), "*Ca*. Methanoproducendum senex" (GMQP bin_32), and "*Ca*. Methylarchaeum tengchongensis" (JZ-2 bin_220). The etymology and descriptions are provided below. The proposed taxonomy of these organisms is summarized in Supplementary Table 3.

"Methanohydrogenotrophicum" (Me.tha.no.hy.dro.gen'o.tro.phi.cum) M. L. n. methanum methane; Gr. n. hydoor, water; Gr. n. gennao, to create; M.L. hydrogenum, hydrogen, that which produces water; N.L. neut. adj. trophicum (from Gr. neut. adj. trophikon) nursing, tending or feeding; N.L. neut. n. hydrogenotrophicum hydrogenotroph; N.L. neut. n. Methanohydrogenotrophicum, methane (-producing) hydrogenotroph. The type species is "*Ca*. Methanohydrogenotrophicum pristinum". "*Ca*. Methanohydrogenotrophicum pristinum" (pris.ti'num) L. masc. adj. pristinum, ancient, venerable. The type material is the metagenomic bin JZ bin_66 (Ga0263245). "Methanogeoarchaeum" (Me.tha.no.ge'o.ar.chae.um) M. L. n. methanum methane; Gr. n. geo Earth; Gr. adj. archeo ancient, also referring to Archaea; Methanogeoarchaeum, methane (-producing) terrestrial archaeon. The type species is "*Ca*. Methanogeoarchaeum hydrogenovorans". "*Ca*. Methanogeoarchaeum hydrogenovorans" (hy.dro.gen.o.vo'rans) M.L. hydrogenum, hydrogen, that which produces water; L. masc. vorus, devouring; L. part. hydrogenovorans, hydrogen-devouring. The type material is the metagenomic bin ZMQR bin_18 (Ga0263256). Phylogenetic analyses support the proposal for a new family inclusive of both "*Ca*. Methanohydrogenotrophicum pristinum" and "*Ca*. Methanogeoarchaeum hydrogenovorans", Methanohydrogenotrophicaceae (Me.tha.no.hy.dro.gen'o.tro.phi.ca'ce.ae) N.L. n. Methanohydrogenotrophicum type genus of the family; L. suff. -aceae ending to denote a family; N.L. fem. pl. n. Methanohydrogenotrophicaceae the family of the genus Methanohydrogenotrophicum.

"Methanomethylovorus" (Me.tha.no.me.thy'lo.vo.rus) M. L. n. methanum methane; methylo, methyl, referring to the methylotrophic methane-producing metabolism; L. masc. vorus, devouring; L. part. Methanomethylovorus, methyl group-devouring methanogen. The type species is "*Ca*. Methanomethylovorus thermophilus". "*Ca*. Methanomethylovorus thermophilus" (ther.mo'phi.lus). Gr. n. therme heat; Gr. adj. philos friendly, loving; N.L. masc. adj. thermophilus heat-

loving. The type material is the metagenomic bin JZ-2 bin_200 (Ga0263249). "Methanomethyloarchaeum" (Me.tha.no.me.thy'lo.ar.chae.um) M. L. n. methanum methane; methylo, methyl, referring to the methylotrophic methane-producing metabolism; Gr. adj. archeo ancient, also referring to Archaea; Methanomethyloarchaeum, methylotrophic methanogen belonging to the Archaea. The type species is "*Ca*. Methanomethyloarchaeum antiquum". "*Ca*. Methanomethyloarchaeum antiquum" (an.ti'.quum) M. L. adj. ancient, referring to the deep-branching position in the Archaea. The type material is the metagenomic bin GMQP bin_44 (Ga0263253). Similarly, two new orders are proposed correspondingly, Methanomethylovorales (Me.tha.no.me.thy'lo.vo.ra'les) N.L. n. Methanomethylovorus type genus of the order, and Methanomethyloarchaeales (Me.tha.no.me.thy'lo.ar.chae'ales) N.L. n. Methanomethyloarchaeum type genus of the order; L. suff. -ales ending to denote an order.

"Hadesarchaeum" (Ha'des.ar.chae.um) M. Gr. n. Greek god of the underworld; Gr. adj. archeo ancient, also referring to Archaea. L. part. Hadesarchaeum, the archaeal god of the underworld. The type species is "*Ca*. Hadesararchaeum tengchongensis". "*Ca*. Hadesararchaeum tengchongensis" (teng.chong.en'sis) originating from Tenghchong, a region of Yunnan Province, China. The type material is the metagenomic bin JZ-2 bin_199 (Ga0263248). "Methanourarchaeum" (Me.tha.no.ur'ar.chae.um) M. L. n. methanum methane; ur, primitive, original, referring to the deeply branching position of the organism in phylogenomic analyses; Gr. adj. archeo ancient, also referring to Archaea. L. part. Methanourarchaeum, the primitive methanogen. The type species is "*Ca*. Methanourarchaum thermotelluricum". "*Ca*. Methanourarchaum thermotelluricum" (ther.mo.tel.lu'ri.cum). Gr. n. therme heat; L. n. telluricus, originating from the Earth; thermotelluricum L. adj. originating from hot Earth. The type material is the metagenomic bin JZ-1 bin_103 (Ga0263246). Phylogenomic analyses support the proposal for a new family inclusive of both "*Ca*. Hadesararchaeum tengchongensis" and "*Ca*. Methanourarchaum thermotelluricum", Hadesarchaeacaceae (Ha.des.ar.ca'ce.ae) N.L. n. Hadesararchaeum type genus of the family; L. suff. -aceae ending to denote a family; N.L. fem. pl. n. Hadesarchaeceae the family of the genus Hadesarchaeum. Similarly, a new order is proposed, Hadesarchaeales (Ha.des.ar.che.a'les) N.L. n. Hadesarchaeum type genus of the order; L. suff. -ales ending to denote an order; N.L. fem. pl. n. Hadesarchaeales the order of the genus Hadesarchaeum. A new class is proposed, Hadesarchaea (Ha.des.ar.ca'.a) N.L. n. Hadesararchaeum type genus of the class; Hadesarchaea the class of the genus Hadesarchaeum. A new phylum is proposed, Hadesarchaeota (Ha.des.ar.ca'.o.ta) N.L. n. Hadesarchaeum type genus of the phylum; L. suff. -ota ending to denote an phylum; N.L. fem. pl. n. Hadesarchaeaota the phylum of the genus Hadesarchaeum.

"Methanomixtatrophicum" (Me.tha.no.mix.ta.tro'phi.cum) M. L. n. methanum methane; M. L. adj. mixed, referring to H₂-dependent methanogenesis; N.L. neut. adj. trophicum (from Gr. neut. adj. trophikon) nursing, tending or feeding; Methanomixtatrophicum, the methanogen using mixed substrates. The type species is "*Ca*. Methanomixtatrophicum sinensis". "*Ca*. Methanomixtatrophicum sinensis" (si.nen'sis). L. adj. originating from China. The type material is the metagenomic bin JZ-2 bin_168 (Ga0263247). Phylogenomic analyses support the proposal for a new family, Methanomixtatrophicaceae (Me.tha.no.mix.ta.tro'phi. ca'ce.ae) N.L. n. Methanomixtatrophicum type genus of the family; L. suff. -aceae ending to denote a family; N.L. fem. pl. n. Methanomixtatrophicaceae the family of the genus Methanomixtatrophicum.

"Methanoproducendum" (Me.tha.no.pro.du.cen'dum) M. L. n. methanum methane; M. L. adj. producer; L. part. Methanoproducendum, the methane producer. The type species is "*Ca*. Methanoproducendum senex". "*Ca*. Methanoproducendum senex" (se'nex). L. n. old man, referring to the deep-branching position of the organism. The type material is the metagenomic bin GMQP bin_32 (Ga0263258). Phylogenomic analyses support the proposal for a new family, Methanoproducendacaceae (Me.tha.no.pro.du.cen'da'ce.ae) N.L. n. Methanoproducendum type genus of the family; L. suff. -aceae ending to denote a family; N.L. fem. pl. n. Methanoproducendacaceae the family of the genus Methanoproducendum.

"Methylarchaeum" (Me.thyl.ar.cha'um) M. L. n. methanum methane; methyl, referring to methylotrophic metabolism; Gr. adj. archeo ancient, also referring to Archaea. L. part. Methylarchaeum, the methane-consuming archaeon. The type species is "*Ca*. Methylarchaeum tengchongensis". "*Ca*. Methylarchaeum tengchongensis" (teng.chong.en'sis) originating from Tengchong, a region of Yunnan Province, China. The type material is the metagenomic bin JZ-2 bin_220 (Ga0263250).

**Reporting summary**. Further information on research design is available in the Nature Research Reporting Summary linked to this article.

## Data availability

The 14 near-complete archaeal genomes are publicly available in the JGI IMG-MER under the Study ID Gs0127627 and WGS accessions Ga0180368 (Unclassified *Nezhaarchaeota* JZ bin_38), Ga0263245 (Unclassified *Nezhaarchaeota* JZ-1 bin_66), Ga0263257 (Unclassified *Nezhaarchaeota* GMQP bin_37), Ga0263256 (Unclassified *Nezhaarchaeota* ZMQR bin_18), Ga0263254 (Unclassified *Verstraetearchaeota* DRTY-6 bin_144), Ga0263253 (Unclassified *Verstraetearchaeota* GMQP bin_44), Ga0263249 (Unclassified *Verstraetearchaeota* JZ-2 bin_200), Ga0263252 (Unclassified

*Verstraetearchaeota* JZ-3 bin_106), Ga0263255 (Unclassified *Verstraetearchaeota* JZ-3 bin_107), Ga0263246 (Unclassified *Hadesarchaeota* JZ-1 bin_103), Ga0263248 (Unclassified *Hadesarchaeota* JZ-2 bin_199), Ga0263247 (Unclassified *Methanomassiliicoccales* JZ-2 bin_168), Ga0263258 (Unclassified *Archaeoglobales* GMQP bin_32), and Ga0263250 (Unclassified *Thaumarchaeota* JZ-2 bin_220). All other relevant data are available upon request.

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

## Acknowledgements

We thank Guangdong Magigene Biotechnology Co., Ltd. China for the assistance in data analysis. This work was financially supported by Science and Technology Infrastructure work project of China Ministry of Science & Technology (No. 2015FY110100), National Natural Science Foundation of China (91951005, 31600298, U1201233 and 31370154), Natural Science Foundation of Guangdong Province, China (No. 2016A030312003), Guangdong Province Science and Technology Innovation Strategy Special Fund (No. 2018B020206001), Guangzhou Municipal People's Livelihood Science and Technology Plan (No. 201803030030), China Postdoctoral Science Foundation (2016M602567), State Key Laboratory of Marine Pollution (SKLMP) Seed Collaborative Research Fund (2019). W.J.L was supported by Guangdong Province Higher Vocational Colleges & Schools Pearl River Scholar Funded Scheme (2014). P.N.E. is supported by an Australian Research Council Discovery Early Career Researcher Award (1700100428). The authors are grateful to the Researchers Supporting Project number (RSP-2019/53), King Saud University, Riyadh, Saudi Arabia.

## Author contributions

Z.S.H., P.N.E., Y.L.W., T.Z., G.W.T., S.W.S., W.H. and W.J.L. conceived the study. Y.N.Q., Y.X.L., Y.L.Q. and Y.Z.R. performed the measurement of physiochemical parameters and DNA extraction. Z.S.H., P.N.E., Y.N.Q., Y.X.L., Y.Z.R., Y.T.C., Y.P.M., J.Y.J. and Y.L.Q. performed the metagenomic analysis, genome binning, functional annotation, and evolutionary analysis. Z.S.H., P.N.E., Y.L.W., W.J.L., G.W.T., M.J.H., K.M.G., and B.P.H. wrote the manuscript. All authors discussed the results and commented on the manuscript.

## Competing interests

The authors declare no competing interests.
