## [Peer Review File · Nature Communications]

Reviewers' comments:

Reviewer #1 (Remarks to the Author):

The paper by Hua et al studied several hot springs in China and expanded the phylogenetic distribution of organisms containing *mcr* to new lineages including Thaumarchaeota and Archaeoglobales. The authors also concluded that *mcr* genes mainly evolved by vertical inheritance, which can support *mcr*-based types of metabolism in the archaeal ancestor.

I found this paper very interesting and just have a few comments and corrections below. I also missed some references to recent literature that is significant especially for the discussion of the evolutionary scenario presented.

- line 140 "...*fpoA-N* complex, minus *fpoEFG* subunits" - Archaeal complex I, called *fpo*, has 11 subunits and is devoid of *fpoEFG*, those subunits do not exist!
In bacterial complex I (*nuo*) the subunits *nuoEFG* are the entry point of electrons from NADH. see for instance Marreiros et al BBA-Bioenergetics 2013. I think a reference for *fpoEFG* appears several times. please correct all, including removing the *fpo*-like from the figure. Replace it with Archaeal complex I or a variation of it.

- The discussion regarding the predicted high OGTs for the common ancestors of methanogens and methanotrophs/alkantrophs is very interesting and goes in line with the Raymann et al PNAS 2015 results, where they calculated a thermophylic ancestor of archaea.

- In addition to Adam et al (ref.1 of the current manuscript), two large scale phylogenetic analysis (Weiss et al Nat. Microbiol. 2016 and Williams et al PNAS 2017) found evidences for the presence of the acetyl-coA complex in LUCA. When addressing the *mcr* complex ancestry, methanogenesis and/or alternative functions of the complex, these results should also be discussed.

- lines 364-365 - also phylogenetic inferences support it(see above)

- lines 373-374 - please provide a brief explanation for serpentinization (and a reference for CH₄ production)

Reviewer #2 (Remarks to the Author):

Hua et al., describe the discovery of several metagenome reconstructed genomes containing *mcr* genes. More specifically, these genes were found deep-branching among others within the

Hadesarchaea, Archaeoglobales or Thaumarchaeota. The authors analyse the metabolism of genomes containing these genes in more detail and try to delineate the evolution of the mcr genes. Overall, the metagenomic analyses are well done and finding the mcr in additionally lineages is interesting, but since these genes have been found outside of the classical methanogenic/methane oxidizers, this finding is not unexpected. My main criticism is regarding the evolutionary aspect of this study. While, this part of the analyses sets the manuscript apart from early work and is very interesting, I do have some issues with the approach taken, which are outlined further below.

Major comments

1. I am not sure if the choice of phylogenetic program (FastTree) is the best tool for deep-evolutionary analyses (even though smaller trees were constructed with different tools but several assumptions were made on the species tree). FastTree is definitely sufficient to place genomes and I do see the advantage with large gene sets, however, a downside is the approximately ML approach that, while providing a speedy analysis, often provides less supported trees (see for example here: <https://www.ncbi.nlm.nih.gov/pmc/articles/PMC5850867/>). Could the authors consider confirming the tree topologies using an alternative tool, such as iqtree that still would allow to run the tree with the full marker set but with a better model, or possibly another tool with a reduced gene set (i.e. RaxML or Bayesian approaches)? I think confirmation the species tree is essential and would support the assumption of the authors of the deep-branching of the new genomes and thus also better support whether their claims on their ancestral metabolism (as discussed for Fig. 1 or Line 115) is indeed well supported (especially, since the authors do not provide a species tree with the numerical support values (see also comments below).

- As a side note: I do have several issues with details of the trees in terms of rooting and reporting of the bootstrapping, which are outlined in more detail in the comments below. However, since the assumptions on the ancestral state of the mcr genes are based are linked to how the trees were rooted, I would suggest to add a) how the tree was rooted and b) based on what evidence was the root chosen.

2. Based on the presented data, I am not completely convinced about the claim that mcr genes were vertically acquired. If I understand this correctly, the authors mainly base this assumption on the comparison of the species with the gene tree. However, at the same time they do admit that "incongruences exist" between species and gene trees (Lines 252-253) and therefore their argument is not very convincing. As an alternative, have the authors investigated the gene origin/gene neighbourhood? I would suggest that the authors more closely investigate the scaffold composition looking at the mcrABG operon and the surrounding genes, i.e. looking at GC content, tetranucleotide frequencies or codon usage..., to confirm their claim.

3. I have not yet used Count myself and while this tool seems to be very good to test gene evolution within related clades, I wonder how well the tool works with more distant/deep-branching lineages? More specifically:

- I imagine that genome completeness could become one issue with ancestral reconstructions. Here, I was wondering if Count accounts for this using a specific algorithm that tries to test whether a gene is lost or simply not part of the genome due to genome completeness or gained due to contamination? While I understand that this might be difficult to implement, it would be helpful if the authors would discuss the limitations of this tool at one point.

- Count seems to mainly use homologous genes for the ancestral reconstruction (and from how I understand it then maps the results of this comparison on the provided species tree). However, this tool does not seem take the phylogenies of individual genes into account. This creates the problem that if any of the individual genes are affected by horizontal gene transfer, duplication events or contamination then this could result in biases in the analyses. More specifically, this could create an

issue with more distant lineages and thus I am not sure if this approach is ideal to understand the gene evolution of the deep-branching lineages or potentially lineages that are more strongly affected by HGT. Therefore, I was wondering if the authors have tested some other tools such as ALE (see also a comparison of the two here: <https://royalsocietypublishing.org/doi/full/10.1098/rstb.2014.0335>)? - Finally, the methods section for the ancestral reconstructions are very short. While the authors do refer to Hua et al., 2018, I would recommend to at least provide some information about the used model type and model parameters.

Other comments

4. Line 105 "on contigs with even sequence depth at medium to high coverage levels, strongly indicating the mcr genes belong to these reconstructed MAGs": Coverage alone is not a strong proof that the contigs belong to the reconstructed MAG. However, since all scaffolds seem to be rather long (and possibly contain even some informative marker genes?) that could be seen as a stronger argument. I would just suggest rewriting this for better clarity.

5. Line 109: What is the used cutoff to define a new order?

6. Line 114-115: How were the 726 genomes selected? More specifically, to be able say in this specific context that these features predate the metabolism of more derived, better known organisms it would be helpful to know how genomes were selected. Since the set of genomes is just a subset of the genomes available at the moment at for example NCBI (or other databases) it might be useful to extend the method section to explain whether the authors made sure that the 726 represent the full archaeal diversity to date. Otherwise one could argue that i.e. excluding diversity for example in the Thaumarchaeota might artificially draw the JZ genome in a deep-branching position. For this specific example, I was also wondering, whether the authors have included Geothermarchaeota, which usually branches close to the Thaumarchaeota?

7. Line 118: 'Surprisingly, these genomes possess previously identified non-mcr-containing Hadesarchaeota genomes'. I do not completely understand the sentence, do the authors mean 'Surprisingly, the previously identified non-mcr-containing Hadesarchaeota genomes'? Please rewrite for better clarity.

8. Line 206: Instead of the citrate synthase I would like to ask, whether this genome encodes for a ATP-citrate lyase as a marker gene for the rTCA? While the citrate synthase might act in reverse, to my understanding this can only be shown by biochemical analyses. Some re-wording might be enough to state this sentence a bit more carefully.

9. Line 277-279: The Thaumarchaeota are on a very separate branch compared to the Proteobacteria (and the position might also depend on how the tree was rooted?), therefore, stating that 'suggesting that aerobic respiration is the result of HGT in these organisms' seems a bit difficult. I would consider removing this sentence and if the authors decide against this, I would suggest to provide the tree file with the support values to be able to better read and judge this figure.

10. Line 306: please specify what is meant by "these organisms" to avoid confusion as the previous sentence refers to both non-mcr and mcr containing Hadesarchaeota.

11. Paragraph starting Line 323: JZ bin_103 and JZ-2 bin_199 show a different structure compared to the other genomes. For example, there are two genes between the mcr operon (what are these genes) and 2 mcrA copies. Could the authors elaborate on what this could mean in terms of the evolutionary

histories of these genes?

12. Line 332: The authors only show an unrooted tree, therefore, no direction of evolution is given. Thus assumptions about the evolution of traditional methanogens should not be drawn from this tree. Therefore, I would suggest to either root the tree (and state how the root was selected) or remove this claim.

13. Line 346: Could the author consider describing, for example in brackets, what CvP bias compares and how this relates to temperature? This is briefly described in the methods but could allow readers for a better understanding without having to go back and forth.

14. Line 350: How was the root set? Especially, since the evolution of mcr is still debated and changing the root would likely change the interpretation of the tree, it would be important to know how this was decided. Can the authors either add this to the methods or add an appropriate reference?

15. Line 417: What are the quality thresholds?

16. Line 423: Please include the version number, especially since there is quite some difference in performances between MetaBat v1 and v2. Additionally, from the text it is not completely clear if the coverage information from all 6 metagenomes was used for the binning of each sample?

17. Line 427: What were the criteria for the manual removal of contigs with contamination?

18. Line 447: With genes less than 122 marker genes, was a cut-off set to remove genomes with too few genes (i.e. less than 50% or something the like?)?

19. Overall the description of the phylogenetic analyses in the methods are very minimal. Overall I would like to ask the authors extend this section a bit and also include some more information in the figures with the trees and the respective captions. More specifically:

- Please add into the figures + describe in the methods the number of trees generated for bootstrapping, the method of bootstrapping and (if applicable) how the trees were rooted.
- Based on some of the trees that show the number of replicates (i.e. Fig S8 with 100), it seems the number is rather low. I would recommend increasing this number or when using Raxml using autoMRE to automatically determine the best number.
- I mention this below but generally it would be helpful to either provide the tree files with the numerical bootstrap supports or indicated a broader range (ie. 70-90, 90-100, 100). For example, showing support values higher than 50% in Fig S8 is not very helpful, as already values below 70 are not very well supported.
- How were the models chosen, did the authors run tests on what the best model is?
- In for example Figure S5 the authors describe a Bayesian phylogeny and briefly discuss this in the methods but without sufficient detail. More specifically, I would recommend adding the following information: how many chains were used, what was the burn-in, how many trees were generated and did the trees converge with the number of trees generated (ideally even reporting some support values for convergence)?

Figures

20. Fig 1:

- Fasttree only produces an approximately-Maximum-Likelihood tree, which is not exactly the same as a maximum-likelihood tree, therefore I would suggest to add the full name here to avoid confusion.

- When using FastTree support values of 70% are already rather low (in another tree values of 50-70 were marked, which are not reliable support values anymore): to allow the reader to better judge the tree I would suggest to either add different color-codes (i.e. for 70-90, 90-100 and 100) or provide the tree file as a supplementary file. The same goes for the tree in Fig S3b or S8.

- Typo: "from a concatenated of 122 proteins", better would be "from a concatenated set of 122 proteins".

21. Fig 2:

- Is there any meaning behind the different color-coding, i.e. in the Wood-Ljungdahl pathway? I guess this could be linked to the taxonomic lineages but it is not completely clear based on the caption, whether this is an aesthetic coloring or linked to the lineages.

- Typo: 'Mathanomassiliicoccales'

22. Fig S2: Typo: "histgrams"

23. Figure S5:

- Typo: "methenogenesis"

- Could the authors indicate the support values in panel a?

- Can the author add to the caption what exactly is collapse in the brown triangle at the top of the tree?

- Panel c: can the authors specify, where exactly the genes were gained or mark this somewhere in the tree or captions?

Supplementary Material

24. There seems to be an issue with Table S2 compared to Figure S2. In Figure S2 there appear to be two genomes with 2 mcrA genes, while only one is listed in Table S2. Additionally, most genomes appear to have a second beta subunit, however, this does not appear to be one the same contig according to Figure S2? Could the authors please clarify this?

25. Line 508: This depends on the journal policies in the end but I would prefer if the authors would also make the metagenomes available, especially since it is unclear what a "reasonable request" should be.

Reviewer #3 (Remarks to the Author):

Title: Insights into the ecological roles and evolution of mcr-containing hot spring Archaea

The study presents a large number of metagenome-assembled genomes (MAGs) from several hot springs in China that contain genes encoding methyl-coenzyme M reductase (MCR) complexes. Among the 14 MAGs, 10 are widely distributed among the TACK superphylum and 4 belong to the Euryarchaeota. Among the former, four seem to represent a new order within Crenarchaeota, five belong to Verstraetearchaeota, and one branches deeply within the Thaumarchaeota. The four genomes from Euryarchaeota belong to the Hadesarchaeota, Methanomassiliicoccales, and Archaeoglobales. The information suggests that the distribution of MCR and MCR-related metabolism is wider among the archaea than previously recognized. Further examination suggests that most MCRs evolved vertically, raising the possibilities that MCR-related metabolism may have existed in the common ancestor of Archaea. The study greatly expands our recognition on the diversity of methane-metabolizing archaea, and provides generality and goes a step further compared to previous studies

on single groups of organisms e.g. Bathyarchaeota. The study should be of broad interest and relates to microbiology, ecology, environmental sciences and evolution. The manuscript in general is well written and was easy to follow. There are some portions where editing is necessary. Some further detailed analyses of the MAGs will better clarify the metabolism of these organisms and should strengthen the manuscript.

Some specific comments for the authors' consideration.

The reviewer is not so sure about the alkane metabolism existing in the common ancestor of Archaea. Can the authors clarify the main observations that suggest so? Is this simply based on the presumption that butane is activated by the MCR in *Syntrophoarchaeum*? Or do the authors also observe the presence of alkylsuccinate synthase homologs on these genomes?

Do the authors have any information on the abundance of mcr-containing organisms within each community? Is there anything worth noting on the communities in general? Although there is the phylogenetic analysis, are there any points worth mentioning or differences among the various mcr loci?

Line 62: origin of these organisms, or origin of mcr genes?

Line 115: more derived, better known organisms; can the authors clarify a bit more here?

Comments for the text on the related metabolism and Fig. 2.

1) beta-oxidation

Fatty acids should join beta-oxidation prior to acyl-CoA in Fig. 2, if they are activated by acyl-CoA synthetases.

Enoyl-CoA, Hydroxyacyl-CoA (if they are not meant to be abbreviated forms)

2) roTCA cycle

What is the node downstream of citrate (Cit)? Taking into account the release of acetyl-CoA, shouldn't it be oxaloacetate (Oxa), or is this acetyl-CoA itself? If it is the latter case, perhaps also connect Cit with Oxa. I also have a question of the ppc reaction. If this is ppc and not pc, this probably represents phosphoenolpyruvate carboxylase (PEPC), and not pyruvate carboxylase. If this is the case, PEP and Oxa, not Pyr and Oxa, should be linked. Consequently, I suggest that the authors present acetyl-CoA diverging from the roTCA cycle, link it with Pyr (por) outside the cycle, and link PEP to Oxa. Nodes in the cycle will decrease from ten to eight. In addition, are there any genomes with ATP-citrate lyase or citryl-CoA lyase genes? Genomes with these genes would confirm a reductive flux.

3) CBB cycle

The lack of the CBB cycle is clearly written in the text, and there is no problem there, but what do the dots indicating *Archaeoglobus*, *Verstraetearchaeota* and *Mathanomassiliicoccales* mean? Slightly confusing, does it represent the presence of RuBisCO? And what is the link between AMP and the cycle?

4) Nucleotide salvage pathway

Many bacteria and archaea use phosphorylases to release ribose moieties from nucleosides. This would result in the generation of ribose 1-phosphate as starting material for PRPP regeneration. The pathway is probably ribose 1-phosphate to ribose 5-phosphate by a phosphomutase and then generation of PRPP. Another point for clarification is the generation of AMP from ATP. Is this correct? Concerning archaeal nucleoside salvage, is there a ribose-1,5-bisphosphate isomerase homolog?

5) Glycolysis and gluconeogenesis

Although this is not discussed and may be outside the main scope of the manuscript, I suggest the authors illustrate the presence/absence of phosphofructokinase (PFK) and fructose-1,6-bisphosphatase (FBPase) in Fig. 2. This would help the readers to immediately recognize whether the Embden-Meyerhof pathway also functions in glycolysis (and energy conservation with sugar

breakdown), and it is not only a pathway for gluconeogenesis (anabolism). Are there sugar transporter genes?

6) Other points

The arrow from CO in the carbon monoxide dehydrogenase reaction should curve right, not left.

Fig. 3A: The colored branches should be described. Does the *Archaeoglobus* branch represent the sequence of *mcrABG* from GMQP_32? Is the *Polytropus marinifundus mcrABG* included in the *Syntrophoarchaeum* branch?

Fig. 3C: Genomes in this study should be indicated with lines with darker color.

Line 379: This sentence needs rephrasing. The summary section should be re-edited for better clarity.

Supplementary figures:

Figure S2: histograms, *mcrABG* in italic twice

Figure S2: Should *Polytropus marinifundus* be included here?

Others

Line 38: ability, respectively.

Line 92: Results and Discussion

Line 121: H₂-dependent

Line 200: remove the first clade

The authors should carefully check the pdf versions of their Tables. Most of the tables were not there (but I eventually found them in the Excel source files)

Responses to the reviewer's comments

Reviewer #1 (Remarks to the Author):

The paper by Hua et al studied several hot springs in China and expanded the phylogenetic distribution of organisms containing *mcr* to new lineages including *Thaumarchaeota* and *Archaeoglobales*. The authors also concluded that *mcr* genes mainly evolved by vertical inheritance, which can support *mcr*-based types of metabolism in the archaeal ancestor.

I found this paper very interesting and just have a few comments and corrections below. I also missed some references to recent literature that is significant especially for the discussion of the evolutionary scenario presented.

Comment 1: line 140 "...*fpoA-N* complex, minus *fpoEFG* subunits" - Archaeal complex I, called *fpo*, has 11 subunits and is devoid of *fpoEFG*, those subunits do not exist!

In bacterial complex I (*nuo*) the subunits *nuoEFG* are the entry point of electrons from NADH. See for instance Marreiros et al BBA-Bioenergetics 2013. I think a reference for *fpoEFG* appears several times. Please correct all, including removing the *fpo*-like from the figure. Replace it with Archaeal complex I or a variation of it.

Response: We thank the reviewer for this comment. The 'fpo' notation has been amended to 'Archaeal complex I' in the main text (Revised Manuscript Lines 157 and 201) and Fig. 2 as suggested.

Comment 2: The discussion regarding the predicted high OGTs for the common ancestors of methanogens and methanotrophs/alkantrophs is very interesting and goes in line with the Raymann et al PNAS 2015 results, where they calculated a thermophylic ancestor of archaea.

Response: Thanks for the positive assessment. As suggested, we referred the recommended paper in the Revised Manuscript (Line 377).

Comment 3: In addition to Adam et al (ref.1 of the current manuscript), two large scale phylogenetic analysis (Weiss et al Nat. Microbiol. 2016 and Williams et al PNAS 2017) found evidences for the presence of the acetyl-coA complex in LUCA. When addressing the *mcr* complex ancestry, methanogenesis and/or alternative functions of the complex, these results should also be discussed.

Response: The acetyl-CoA synthase/carbon monoxide dehydrogenase been discussed in context of the widespread nature of the *mcr* complex recently found across the archaea, suggesting that these complexes are equally key in the metabolism of early ancestors in the Archaea (see Revised Manuscript Lines 339-342).

Comment 4: lines 364-365 - also phylogenetic inferences support it (see above)

Response: We agree with the reviewer, the papers of Weiss et al. (2016) and Williams et al. (2017) mentioned above have been added (see Revised Manuscript Line 391).

Comment 5: lines 373-374 - please provide a brief explanation for serpentinization (and a reference for
CH₄ production)

Response: Done as suggested (see Revised Manuscript, Line 405-407)

Reviewer #2 (Remarks to the Author):

Hua et al., describe the discovery of several metagenome reconstructed genomes containing *mcr* genes.
More specifically, these genes were found deep-branching among others within the *Hadesarchaea*,
*Archaeoglobales* or *Thaumarchaeota*. The authors analyze the metabolism of genomes containing these
genes in more detail and try to delineate the evolution of the *mcr* genes. Overall, the metagenomic
analyses are well done and finding the *mcr* in additionally lineages is interesting, but since these genes
have been found outside of the classical methanogenic/methane oxidizers, this finding is not
unexpected. My main criticism is regarding the evolutionary aspect of this study. While, this part of the
analyses sets the manuscript apart from early work and is very interesting, I do have some issues with
the approach taken, which are outlined further below.

Major comments

Comment 1: I am not sure if the choice of phylogenetic program (FastTree) is the best tool for
deep-evolutionary analyses (even though smaller trees were constructed with different tools but several
assumptions were made on the species tree). FastTree is definitely sufficient to place genomes and I do
see the advantage with large gene sets, however, a downside is the approximately ML approach that,
while providing a speedy analysis, often provides less supported trees (see for example
here: <https://www.ncbi.nlm.nih.gov/pmc/articles/PMC5850867/>). Could the authors consider
confirming the tree topologies using an alternative tool, such as iqtree that still would allow to run the
tree with the full marker set but with a better model, or possibly another tool with a reduced gene set
(i.e. RaxML or Bayesian approaches)?

Response: We agree with the reviewer that while it might be reliable to use FastTree to construct an
approximate ML tree, potential problems exist with genome placement. We accept the reviewer's
concerns and in the revised Fig. 1 we use IQ-TREE to reconstruct the phylogeny of the Archaea.

I think confirmation the species tree is essential and would support the assumption of the authors of the
deep-branching of the new genomes and thus also better support whether their claims on their ancestral
metabolism (as discussed for Fig. 1 or Line 115) is indeed well supported (especially, since the authors
do not provide a species tree with the numerical support values (see also comments below).

Response: Further, to the improved taxonomic assignment using iqtree we have included several
recently published novel *mcr*-containing lineages in addition to those novel *mcr*-containing organisms
that we report in this current paper. These additions firstly give a more complete understanding these
novel archaeal lineages and provide complete summarization of currently published *mcr*-containing
taxa. Also, it provides further evidence to consolidate our claim that the newly reported thermophilic
microbes are located basal in several different lineages. This result further strengthens our assertion of

an ancestral metabolism is associated with methanogenesis or methane oxidation.

-As a side note: I do have several issues with details of the trees in terms of rooting and reporting of the
bootstrapping, which are outlined in more detail in the comments below. However, since the
assumptions on the ancestral state of the *mcr* genes are based are linked to how the trees were rooted, I
would suggest to add a) how the tree was rooted and b) based on what evidence was the root chosen.

Response: We agree that the bootstrap values are required and now have been included. Also, we have
rooted the trees based on the assumptions that TACK lineages are evolutionarily more ancient than
euryarchaeal lineages, rather than rooting based on the proposed alkane oxidizing Mcr sequences as
these sequences are likely the results of gene duplication only within the *Euryarcheota*. A paragraph
has been included to state these points (lines 282-286)

Comment 2: Based on the presented data, I am not completely convinced about the claim that *mcr*
genes were vertically acquired. If I understand this correctly, the authors mainly base this assumption
on the comparison of the species with the gene tree. However, at the same time they do admit that
“incongruences exist” between species and gene trees (Lines 252-253) and therefore their argument is
not very convincing.

Response: We believe the for the most of the *mcr* is transferred by vertical decent, but do agree that
there are incongruences in the taxonomy which leads to the statement in the abstract that we ‘suggest a
mostly vertical evolution for these genes’ from observations of the mostly similar topologies between
the concatenated *mcrABG* gene and archaeal species trees. Potential instances of HGT have been noted
in the Revised Manuscript (Lines 286-288 and 299-300).

As an alternative, have the authors investigated the gene origin/gene neighborhood? I would suggest
that the authors more closely investigate the scaffold composition looking at the *mcrABG* operon and
the surrounding genes, i.e. looking at GC content, tetranucleotide frequencies or codon usage..., to
confirm their claim.

As suggested, in the revised manuscript, the gene clusters nearby the *mcr*-complex were investigated to
detect the potential HGTs. However, in the corresponding scaffolds, we observed nearly no significant
GC bias of *mcr*-complex to their neighbors (Table S3). Also, many nearby genes were identified as
being either methanogenesis related marker genes and/or genes related to energy conservation. These
genes always span > 30 kbp on individual contigs, which suggests a low possibility of acquisition via a
single HGT event (Pang et al., 2019). Overall, we think that all evidence likely confirms our claim that
mostly vertical decent, rather than mostly HGT (limited examples include *Archaeoglobi* and
*Bathyarchaeota mcr* genes), dominates *mcrABG* gene acquisition (see Revised Manuscript Lines
273-382).

Comment 3: (1) I have not yet used Count myself and while this tool seems to be very good to test gene
evolution within related clades, I wonder how well the tool works with more distant/deep-branching
lineages? More specifically:

Response: We agree COUNT is a very good for gene evolution testing within related clades, we also
suggest that COUNT is useful for the inference of ancestral gene contents of distant lineages, such as,
the new class *Methanonatronarchaeia* which deeply branched near Halobacteriales (Sorokin et al.,
2017), the whole archaea domain (Wolf et al., 2012), even the divergent giant viruses (Schulz et al.,
2017). Furthermore, the authors who developed the tool also used it for the inference of gains and
losses of the whole archaea domain (Csűrös and Miklós, 2009) using the same algorithm to infer the
gene content evolution of archaea was subsequently used to develop COUNT (Csüös, 2010).

-(2) I imagine that genome completeness could become one issue with ancestral reconstructions. Here,
I was wondering if Count accounts for this using a specific algorithm that tries to test whether a gene is
lost or simply not part of the genome due to genome completeness or gained due to contamination?
While I understand that this might be difficult to implement, it would be helpful if the authors would
discuss the limitations of this tool at one point.

Response: We agree, MAG completeness is an important issue which may affect the accuracy of
evolutionary history inference. In our previous paper (Hua et al., 2018), the same question was raised
by a reviewer about COUNT with respect to the evolution of Thaumarchaeota and Aigarchaeota phyla.
For this we showed that comparisons between genomes with completeness > 80% and >90% produced
no significant differences between different selection criteria for the inference of gains and losses of
internal nodes. At this higher completeness the results may be biased because of overestimation of gene
families in extant genomes. Also, at this higher stringency, some genomes (and nodes) were lost
leading to the lower resolution of evolutionary histories for specific lineages. Therefore, we used lower
completeness criteria to recruit more genomes to better understand their evolutionary histories but had
to balance the bias of incomplete MAGs by focusing on gain events rather than gene losses.

-(3) Count seems to mainly use homologous genes for the ancestral reconstruction (and from how I
understand it then maps the results of this comparison on the provided species tree). However, this tool
does not seem take the phylogenies of individual genes into account. This creates the problem that if
any of the individual genes are affected by horizontal gene transfer, duplication events or
contamination then this could result in biases in the analyses. More specifically, this could create an
issue with more distant lineages and thus I am not sure if this approach is ideal to understand the gene
evolution of the deep-branching lineages or potentially lineages that are more strongly affected by HGT.
Therefore, I was wondering if the authors have tested some other tools such as ALE (see also a
comparison of the two here: <https://royalsocietypublishing.org/doi/full/10.1098/rstb.2014.0335>)?

Response: While we concede that ALE (and others including Prunier and AnGST) could detect HGT,
we believe that they are unsuitable in this case as we are identifying the succession of specific lineages
by detecting key gene events (gains and losses) across time based on parsimony inference. As the main
point of the associated analysis is to uncover the evolutionary histories of *mcr*-complex and we propose
this is mostly the result of vertical decent for *mcrABG* genes by comparing *mcr* gene tree and
phylogenomic tree (species tree) directly, even though HGTs do exist. ALE may help us better
understand HGT, duplication and loss, in shaping the genetic diversity of those lineages. However, it is
likely it would lead to a more complicated scenario since a single gain event of one gene family at one
node might contain several HGT/duplication events.

- (4) Finally, the methods section for the ancestral reconstructions are very short. While the authors do
refer to Hua et al., 2018, I would recommend to at least provide some information about the used
model type and model parameters.

Response: Revised as suggested (see Revised Manuscript Lines 525-530).

Other comments

Comment 4: Line 105 “on contigs with even sequence depth at medium to high coverage levels,
strongly indicating the *mcr* genes belong to these reconstructed MAGs”: Coverage alone is not an
strong proof that the contigs belong to the reconstructed MAG. However, since all scaffolds seem to be
rather long (and possibly contain even some informative marker genes?) that could be seen as a
stronger argument. I would just suggest rewriting this for better clarity.

Response: Revised as suggested. As said, the evenness of coverage only reflects the accuracy of
assembled *mcr*-containing scaffolds, rather than evidence that scaffolds belong to the reconstructed
MAGs. On the contigs containing *mcr* genes we do observe several collocated archaeal tRNAs,
chaperone proteins and/or ribosomal proteins (see Revised Fig. S3). Also, those *mcr*-containing
scaffolds are always large with lengths of > 50 kbp, with eight of the 14 on contigs that are greater than
100 kbp. An amended description has been added to the revised Manuscript (Lines 115-120).

Comment 5: Line 109: What is the used cutoff to define a new order?

Response: The cutoff was used as per GTDB taxonomy Relative Evolutionary Distance (RED) scores
for order level taxonomy (Parks et al. 2018), this has been mentioned in the Revised Manuscript (Lines
481-483).

Comment 6: Line 114-115: How were the 726 genomes selected? More specifically, to be able say in
this specific context that these features predate the metabolism of more derived, better known
organisms it would be helpful to know how genomes were selected. Since the set of genomes is just a
subset of the genomes available at the moment at for example NCBI (or other databases) it might be
useful to extend the method section to explain whether the authors made sure that the 726 represent the
full archaeal diversity to date. Otherwise one could argue that i.e. excluding diversity for example in
the Thaumarchaeota might artificially draw the JZ genome in a deep-branching position. For this
specific example, I was also wondering, whether the authors have included Geothermarchaeota, which
usually branches close to the Thaumarchaeota?

Response: The new generated phylogeny has been constructed based on 892 MAGs/genomes to reflect
the increased number of genomes since this analysis was first performed (on 726 MAGs/genomes),
including the 14 from this study. These genomes were selected from an archaeal dereplicated NCBI
MAGs/genomes that met a quality score of greater than 70% using CheckM (‘medium to high quality’),
MAGs/genomes and had an alignment greater than 50% of amino acids compared to a full alignment to
reliably place the MAGs/genomes in the species tree (see Revised Manuscript Lines 478-482). More

neighbors inside/outside of *Thaumarchaeota* MAG in this study have been added, including the
*Thaumarchaeota*-belonging MAG OP bin 015 from a recent study (Berghuis et al., 2019) and four
MAGs from *Caldiarchaem* which located outside of *Thaumarchaeota*. Evidences still support the
placement of this MAG into Thaumarchaeota.

Comment 7: Line 118: ‘Surprisingly, these genomes possess previously identified non-mcr-containing
Hadesarchaeota genomes’. I do not completely understand the sentence, do the authors mean
‘Surprisingly, the previously identified non-mcr-containing Hadesarchaeota genomes’? Please rewrite
for better clarity.

Response: Correct. We do mean that previous reported Hadesarchaeota are lacking *mcr* genes. Revised
as suggested (see Revised Manuscript Line 211).

Comment 8: Line 206: Instead of the citrate synthase I would like to ask, whether this genome encodes
for a ATP-citrate lyase as a marker gene for the rTCA? While the citrate synthase might act in reverse,
to my understanding this can only be shown by biochemical analyses. Some re-wording might be
enough to state this sentence a bit more carefully.

Response: No ATP-citrate lyases and citryl-CoA lyases were detected in the *Archaeoglobales*
GMPQ_32 MAG or any of 14 *mcr*-containing MAGs. In terms of citrate synthase, strong experimental
evidence has been presented for thermophiles such as *Thermosulfidibacter takaii* and *Desulfurella*
*acetivorans* showing that they were able to utilize citrate for the fixation of carbon dioxide via the
roTCA cycle (Mall et al., 2018; Nunoura et al., 2018). This sentence has been reworded for better
clarification (see Revised Manuscript Lines 236-238).

Comment 9: Line 277-279: The Thaumarchaeota are on a very separate branch compared to the
Proteobacteria (and the position might also depend on how the tree was rooted?), therefore, stating that
‘suggesting that aerobic respiration is the result of HGT in these organisms’ seems a bit difficult. I
would consider removing this sentence and if the authors decide against this, I would suggest providing
the tree file with the support values to be able to better read and judge this figure.

Response: Revised as suggested. These two sentences associated with the phylogeny of *coxB* have been
removed, as well as references to these genes in Supplementary Information Fig. S6.

Comment 10: Line 306: please specify what is meant by “these organisms” to avoid confusion as the
previous sentence refers to both non-*mcr* and *mcr* containing Hadesarchaeota.

Response: Here we mean the *mcr*-containing Hadesarchaeota. This sentence was reworded to avoid the
ambiguity (see Revised Manuscript Line 326).

Comment 11: Paragraph starting Line 323: JZ bin_103 and JZ-2 bin_199 show a different structure
compared to the other genomes. For example, there are two genes between the *mcr* operon (what are
these genes) and 2 *mcrA* copies. Could the authors elaborate on what this could mean in terms of the
evolutionary histories of these genes?

Response: Unfortunately, the genes between *mcr* genes are hypothetical and cannot be identified. We do however believe the two *mcrA* genes are likely a relatively recent gene duplication event and neofunctionalisation as seen in the alkane oxidizing *mcr* genes of Syntrophoarchaeum as suggested by Evans et al. (2019). This has been mentioned in the text that there are two copies of the *mcrA* genes likely as a result of gene duplication (Revised Manuscript Lines 359-361).

Comment 12: Line 332: The authors only show an unrooted tree, therefore, no direction of evolution is given. Thus, assumptions about the evolution of traditional methanogens should not be drawn from this tree. Therefore, I would suggest to either root the tree (and state how the root was selected) or remove this claim.

Response: We have rooted the trees based on the assumptions that TACK lineages are evolutionarily more ancient than euryarchaeal lineages, rather than rooting based on the proposed alkane oxidizing Mcr sequences as these sequences are likely the results of gene duplication only within the Euryarchaeota. A paragraph has been included to state these points (lines 282-286).

Comment 13: Line 346: Could the author consider describing, for example in brackets, what CvP bias compares and how this relates to temperature? This is briefly described in the methods but could allow readers for a better understanding without having to go back and forth.

Response: Revised as suggested. In brief, CvP bias means the difference between proportions of charged and polar amino acids. Pearson's correlation was conducted between the CvP biases and OGTs of cultured *mcr*-containing microorganisms. Significant positive correlation was observed and has been revised in the manuscript (see Revised Manuscript Line 370).

Comment 14: Line 350: How was the root set? Especially, since the evolution of *mcr* is still debated and changing the root would likely change the interpretation of the tree, it would be important to know how this was decided. Can the authors either add this to the methods or add an appropriate reference?

Response: We have rooted the trees based on the assumptions that TACK lineages are evolutionarily more ancient than euryarchaeal lineages, rather than rooting based on the proposed alkane oxidizing Mcr sequences as these sequences are likely the results of gene duplication only within the Euryarchaeota. A paragraph has been included to state these points (lines 282-286). Regarding to the ancestral sequence reconstruction for ancestor nodes, the input tree for PAML is unrooted under the condition of no clock (clock=0). Therefore, no matter where the root is, the ancestral sequences and OGT estimations will not change.

Comment 15: Line 417: What are the quality thresholds?

Response: The quality control steps included refinement of contamination in bins and removal of low quality of raw sequence reads are described in detail in Revised Methods (see Revised Manuscript Lines 447-451).

Comment 16: Line 423: Please include the version number, especially since there is quite some
difference in performances between MetaBat v1 and v2. Additionally, from the text it is not completely
clear if the coverage information from all 6 metagenomes was used for the binning of each sample?

Response: Thank you for the suggestion. In the Revised Methods, we have added the version number
for MetaBat and other tools if possible. In our study, we used MetaBat (v2.12.1) for genome binning.
Coverage was only used for binning individual samples and was independent of other samples (i.e.
co-assembly was not used), (see revised Manuscript lines 457-458).

Comment 17: Line 427: What were the criteria for the manual removal of contigs with contamination?

Response: Scaffolds with abnormal coverage information or discordant positions in ESOM were
removed (Revised Manuscript Line 460). Some contamination remained and could not be removed
based on CheckM results, these contigs were often small and did not contain genes critical to the
current analyses of metabolism.

Comment 18: Line 447: With genes less than 122 marker genes, was a cut-off set to remove genomes
with too few genes (i.e. less than 50% or something the like?)?

Response: All genomes (both *mcr* and non *mcr*-containing Archaea) were subjected to CheckM (Parks
et al. 2015) quality metric scores of greater than 70% which are classified as 'quality genomes' were
applied to ensure the integrity of analyses (also see comments about the 726 MAGs/genome
taxonomy).

Comment 19: Overall the description of the phylogenetic analyses in the methods are very minimal.
Overall I would like to ask the authors extend this section a bit and also include some more information
in the figures with the trees and the respective captions. More specifically:

-Please add into the figures + describe in the methods the number of trees generated for bootstrapping,
the method of bootstrapping and (if applicable) how the trees were rooted.

Response: Revised as suggested. Previously we used Fasttree to reconstruct the phylogenomic tree and
RAxML to reconstruct the phylogenies of concatenated and individual *mcr*ABG genes. In the Revised
Manuscript, we change to IQ-tree to reconstruct all the phylogenies mentioned above with ultrafast
bootstrapping and SH-aLRT tests. Each test was repeated 1000 times (see Revised Manuscript Lines
488-490).

-Based on some of the trees that show the number of replicates (i.e. Fig S8 with 100), it seems the
number is rather low. I would recommend increasing this number or when using Raxml using
autoMRE to automatically determine the best number.

Response: As mentioned above, we have conducted ultrafast bootstrapping and SH-aLRT tests with
1000 replicates for the new generated phylogenies.

-I mention this below but generally it would be helpful to either provide the tree files with the

numerical bootstrap supports or indicated a broader range (ie. 70-90, 90-100, 100). For example,
showing support values higher than 50% in Fig S8 is not very helpful, as already values below 70 are
not very well supported.

Response: Revised as suggested. In Revised Fig 1, 3, and S8, bootstrap values > 70 were shown on the
trees. Also, we have supplied the raw newick trees to the supplementary materials (Supplementary Data
1 and 2)

-How were the models chosen, did the authors run tests on what the best model is?

Response: For the new phylogenies mentioned above, we use ModelFinder (Kalyaanamoorthy et al.,
2017) to decide the best model (see Revised Manuscript Line 490).

-In for example Figure S5 the authors describe a Bayesian phylogeny and briefly discuss this in the
methods but without sufficient detail. More specifically, I would recommend adding the following
information: how many chains were used, what was the burn-in, how many trees were generated and
did the trees converge with the number of trees generated (ideally even reporting some support values
for convergence)?

Response: In terms of Bayesian tree constructed by MrBayes (Ronquist et al., 2012), two independent
runs with four chains of each were run. For each chain, different lineages start with different
generations with the same burn-in fraction of 0.25 was conducted. Detailed information related to
Bayesian-based phylogenetic tree construction was described in Methods (see Revised Manuscript
Lines 543-549).

Figures

Comment 20: Fig 1:

-(a) Fasttree only produces an approximately-Maximum-Likelihood tree, which is not exactly the same
as a maximum-likelihood tree, therefore I would suggest to add the full name here to avoid confusion.

Response: Yes, we agree that Fasttree can be limited in constructing phylogeny. In conjunction with
reviewer 2's suggestions we have opted to use IQ-tree to rebuild those trees in the Revised Manuscript.

-(b) When using FastTree support values of 70% are already rather low (in another tree values of 50-70
were marked, which are not reliable support values anymore): to allow the reader to better judge the
tree I would suggest to either add different color-codes (i.e. for 70-90, 90-100 and 100) or provide the
tree file as a supplementary file. The same goes for the tree in Fig S3b or S8.

Response: As suggested, in Figure 1 and S8, only nodes with support values > 70 are shown. In
Revised Figure S4b (the same as previous Fig. 3b), nodes with confidence of > 80% are shown as a
measure of high quality in the Bayesian tree. Also, we have generated newick trees for Figures 1 and 3a
that have been supplied as supplementary data.

-(c) Typo: "from a concatenated of 122 proteins", better would be "from a concatenated set of 122

proteins”.

Response: Revised as suggested.

Comment 21: Fig 2:

-Is there any meaning behind the different color-coding, i.e. in the Wood-Ljungdahl pathway? I guess
this could be linked to the taxonomic lineages, but it is not completely clear based on the caption,
whether this is an aesthetic coloring or linked to the lineages.

-Typo: ‘Mathanomassiliicoccales’

Response: Yes, this is an aesthetic coloring and different colors are used for distinguishing different
metabolic modules (see revised caption for Figure 2). Typo was revised as suggested.

Comment 22: Fig S2: Typo: “histgrams”

Response: Revised as suggested.

Comment 23: Figure S5:

(a) Typo: “methenogenesis”

Response: Revised as suggested.

(b) Could the authors indicate the support values in panel a?

Response: We have added a sentence in caption to describe this, since all nodes seen on the phylogeny
are in high confidence with posterior probabilities equal to 1.

(c) Can the author add to the caption what exactly is collapse in the brown triangle at the top of the tree?

Response: Detailed group information has been recorded in Supplementary Data 6.

(d) Panel c: can the authors specify, where exactly the genes were gained or mark this somewhere in
the tree or captions?

Response: All genes in orange color are gained by Unclassified Thaumarchaeota sp. JZ-2 bin_220.
Related description has been added in caption.

Supplementary Material

Comment 24: There seems to be an issue with Table S2 compared to Figure S2. In Figure S2 there
appear to be two genomes with 2 *mcrA* genes, while only one is listed in Table S2. Additionally, most
genomes appear to have a second beta subunit, however, this does not appear to be one the same contig
according to Figure S2? Could the authors please clarify this?

Response: Yes, there was an issue with the MAGs and numbers of *mcrA* and *mcrB* genes. Several
*mcrA2* genes were incorrectly assigned as *mcrB*, this error has now been corrected. A copy paste error
resulted in only one Hadesarchaeota genome showing that 2 *mcrA* gene present. This result has been
corrected in Supplementary Data 3 (the same as previous Table S2) to show that JZ-2 bin_199 and JZ-1
bin_103 both have 2 *mcrA* copies.

Comment 25: Line 508: This depends on the journal policies in the end but I would prefer if the authors
would also make the metagenomes available, especially since it is unclear what a “reasonable request”
should be.

Response: We will make available the assembled genomes used in the analysis immediately and raw
data soon after. The reason for delaying the submission of the raw data is that other researchers used
our metagenomes to publish on the *mcr* containing organisms after we had to publish the metagenomes
for an earlier paper (Hua et al. 2018), we are concerned that novel data in manuscripts currently under
development will again be compromised.

Reviewer #3 (Remarks to the Author):

Title: Insights into the ecological roles and evolution of *mcr*-containing hot spring Archaea
The study presents a large number of metagenome-assembled genomes (MAGs) from several hot
springs in China that contain genes encoding methyl-coenzyme M reductase (MCR) complexes.
Among the 14 MAGs, 10 are widely distributed among the TACK superphylum and 4 belong to the
*Euryarchaeota*. Among the former, four seem to represent a new order within *Crenarchaeota*, five
belong to Verstraetearchaeota, and one branches deeply within the *Thaumarchaeota*. The four genomes
from *Euryarchaeota* belong to the *Hadesarchaeota*, *Methanomassiliicoccales*, and *Archaeoglobales*.
The information suggests that the distribution of MCR and MCR-related metabolism is wider among
the archaea than previously recognized. Further examination suggests that most MCRs evolved
vertically, raising the possibilities that MCR-related metabolism may have existed in the common
ancestor of Archaea. The study greatly expands our recognition on the diversity of
methane-metabolizing archaea and provides generality and goes a step further compared to previous
studies on single groups of organisms e.g. *Bathyarchaeota*. The study should be of broad interest and
relates to microbiology, ecology, environmental sciences and evolution. The manuscript in general is
well written and was easy to follow. There are some portions where editing is necessary. Some further
detailed analyses of the MAGs will better clarify the metabolism of these organisms and should
strengthen the manuscript.

Some specific comments for the authors' consideration.

Comment 1: The reviewer is not so sure about the alkane metabolism existing in the common ancestor
of Archaea. Can the authors clarify the main observations that suggest so? Is this simply based on the
presumption that butane is activated by the MCR in *Syntrophoarchaeum*? Or do the authors also
observe the presence of alkylsuccinate synthase homologs on these genomes?

Response: After the addition of the recently available *mcr*-containing MAGs it became apparent that
alkane oxidation is present only in the *Euryarchaeota* lineages, with the exception of the

Bathyarchaeota (which could be explained by HGT). Therefore it is unlikely that alkane oxidation was
present in an early ancestor, therefore we have rooted trees using the TACK lineages as they have been
suggested to be more ancient than the *Euryarchaeota* lineages (see Revised Manuscript Lines 295-301).
We did find alkylsuccinate synthase genes in *mcr*-containing MAGs, but not in ‘alkane oxidizing’
MAGs. Also, inferences that alkane metabolism (butane/propane in *Ca. Syntrophoarchaeum* and ethane
in the *Argoarchaeum* lineages) has now been found in two lineages that span several clades of *Mcr*
sequences (Laso-Perez et al. 2016 and Chen et al. 2019), but they are still restricted to the
*Euryarchaeota*.

Comment 2: Do the authors have any information on the abundance of *mcr*-containing organisms
within each community? Is there anything worth noting on the communities in general?

Response: Generally, all *mcr*-containing bins show low abundances in corresponding communities with
relative abundances < 1% (see Revised Table 1).

Although there is the phylogenetic analysis, are there any points worth mentioning or differences
among the various *mcr* loci?

Response: There are interesting points in the case of *Archaeoglobi* and *Hadesarchaeota*, this
information has now been added to the Revised Manuscript (Lines 186-193 and 215-219)

Comment 3: Line 62: origin of these organisms, or origin of *mcr* genes?

Response: Correct. Revised as suggested (see Revised Manuscript Line 71).

Comment 4: Line 115: more derived, better known organisms; can the authors clarify a bit more here?

Response: Here the “more derived, better known organisms” means the traditional *mcr*-containing
microbes which belong to phylum *Euryarchaeota* (see Revised Manuscript Lines 131-132).

Comment 5: Comments for the text on the related metabolism and Fig. 2.

1) beta-oxidation

Fatty acids should join beta-oxidation prior to acyl-CoA in Fig. 2, if they are activated by acyl-CoA
synthetases.

Enoyl-CoA, Hydroxyacyl-CoA (if they are not meant to be abbreviated forms)

Response: Revised as suggested. In our MAGs, all but one contain one or more genes encoding
acyl-CoA synthetase (K01897). As mentioned, acyl-CoA synthetase is used for the activation of fatty
acids. The generated acyl-CoA then enters the catabolic process. Also, the abbreviations including
Enyl-CoA, Hydroxyacyl-CoA, and Ketoacyl-CoA have been changed (see Revised Figure 2).

2) roTCA cycle

What is the node downstream of citrate (Cit)? Taking into account the release of acetyl-CoA, shouldn't
it be oxaloacetate (Oxa), or is this acetyl-CoA itself? If it is the latter case, perhaps also connect Cit

with Oxa. I also have a question of the ppc reaction. If this is ppc and not pc, this probably represents
phosphoenolpyruvate carboxylase (PEPC), and not pyruvate carboxylase. If this is the case, PEP and
Oxa, not Pyr and Oxa, should be linked. Consequently, I suggest that the authors present acetyl-CoA
diverging from the roTCA cycle, link it with Pyr (por) outside the cycle, and link PEP to Oxa. Nodes in
the cycle will decrease from ten to eight. In addition, are there any genomes with ATP-citrate lyase or
citryl-CoA lyase genes? Genomes with these genes would confirm a reductive flux.

Response: Thanks for such a detailed explanation. No ATP-citrate lyase and citryl-CoA lyase genes
were detected in any MAGs. It was well documented that the roTCA cycle could utilize citrate synthase
(CS) to cleave citrate into oxaloacetate and acetyl-CoA and then fix carbon dioxide (Mall et al., 2018;
Nunoura et al., 2018). Additional check shows that our MAGs contain not only the *ppc*
(phosphoenolpyruvate carboxylase), but also *pc* (pyruvate carboxylase). It is correct about the
descriptions of the conversions among different substrates. These changes have been made to Figure 2
in combination with the suggestion of reviewer's comments.

3) CBB cycle

The lack of the CBB cycle is clearly written in the text, and there is no problem there, but what do the
dots indicating *Archaeoglobus*, *Ca. Verstraetearchaeota* and *Mathanomassiliicoccales* mean? Slightly
confusing, does it represent the presence of RuBisCO? And what is the link between AMP and the
cycle?

Response: Yes, this information is not clear. These genomes surely do not possess a complete CBB
cycle since all of them lack the *prkB* gene. But we do observe RuBisCO genes in several of the
genomes, suggesting the potential function in a nucleoside-salvaging pathway. In Revised Figure 2, we
have moved those dots to below the *rbcL* gene (see revised Figure 2). Also, the linking of AMP to CBB
is incorrect, this has been corrected to link Ribose-5P to the CBB cycle (see revised Figure 2).

4) Nucleotide salvage pathway

Many bacteria and archaea use phosphorylases to release ribose moieties from nucleosides. This would
result in the generation of ribose 1-phosphate as starting material for PRPP regeneration. The pathway
is probably ribose 1-phosphate to ribose 5-phosphate by a phosphomutase and then generation of PRPP.
Another point for clarification is the generation of AMP from ATP. Is this correct? Concerning archaeal
nucleoside salvage, is there a ribose-1,5-bisphosphate isomerase homolog?

Response: Yes, several genes in those MAGs were identified as ribose-1,5-bisphosphate isomerase,
which means a complete nucleotide salvage pathway from AMP to Glycerate-3P is present. The
previous Figure 2 was incorrect for the generation of AMP from ATP. Actually, what ATP point to is
ADP. In the revised version, we have removed this due to the limited space. Also, we failed to detect
ribose phosphomutase, but instead observed phosphomannomutase / phosphoglucomutase (K15778)
genes which cannot convert ribose 5P from ribose 1P. Alternatively, ribose generated may utilize
ribokinase and ribose-phosphate pyrophosphokinase to perform the conversion of ribose to ribose-5p
and then PRPP. The Fig. 2 and Table S2 were revised accordingly.

5) Glycolysis and gluconeogenesis

Although this is not discussed and may be outside the main scope of the manuscript, I suggest the
authors illustrate the presence/absence of phosphofructokinase (PFK) and fructose-1,6-bisphosphatase
(FBPase) in Fig. 2. This would help the readers to immediately recognize whether the
Embden-Meyerhof pathway also functions in glycolysis (and energy conservation with sugar
breakdown), and it is not only a pathway for gluconeogenesis (anabolism). Are there sugar transporter
genes?

Response: From the analyses of the genomes we detected neither the phosphofructokinase nor
2-dehydro-3-deoxy-D-gluconate aldolase in any genome, indicating these *mcr*-containing microbes
cannot conserve energy via either EMP or ED glycolysis pathway. Instead, we observed the occurrence
of fructose-1,6-bisphosphatase (FBP, K01622) which suggests ability for gluconeogenesis. Here, we
changed the pathway name from “glycolysis” to “gluconeogenesis” and labelled the FBP to indicate
that this pathway is anabolic. No Phosphotransferase sugar transporters were observed, only those
associated with single sugar, lipooligosaccharide transporter or unclassified transporters (K02056 and
K02057).

6) Other points

The arrow from CO in the carbon monoxide dehydrogenase reaction should curve right, not left.

Response: Revised as suggested.

Comment 6: Fig. 3A: The colored branches should be described. Does the *Archaeoglobus* branch
represent the sequence of *mcrABG* from GMQP_32? Is the *Polytropus marinifundus mcrABG* included
in the *Syntrophoarchaeum* branch?

Response: The representative for the *Archaeoglobus* branch in previous Fig. 3A is GMQP_32 from this
study. In revised Fig. 3a, *Polytropus marinifundus* (Boyd et al., 2019) and other MAGs from
*Archaeoglobi* are also included (Wang et al. 2019).

Fig. 3C: Genomes in this study should be indicated with lines with darker color.

Response: In the revised version of Fig. 3c, genomes in this study were highlighted in red.

Comment 7: Line 379: This sentence needs rephrasing. The summary section should be re-edited for
better clarity.

Response: This section has been revised to reflect the reviewer’s comments (see Revised Manuscript
Lines 410-421).

Supplementary figures:

Comment 8: Figure S2: histograms, *mcrABG* in italic twice

Response: Revised as suggested.

Comment 9: Figure S3: Should *Polytropus marinifundus* be included here?

Response: Thanks for the suggestion. We have added *Polytropus marinifundus* to the revised Figure S4
(another figure has been added before it).

Others

Comment 10: Line 38: ability, respectively.

Response: The description related to *Archaeoglobales* has been removed since a recent study reported
this finding.

Comment 11: Line 92: Results and Discussion

Response: Revised as suggested.

Comment 12: Line 121: H2-dependent

Response: Revised as suggested.

Comment 13: Line 200: remove the first clade

Response: Revised as suggested.

Comment 14: The authors should carefully check the pdf versions of their Tables. Most of the tables
were not there (but I eventually found them in the Excel source files)

Response: These have been included to reflect the reviewer's comments.

**References**

Berghuis, B. A., et al. Hydrogenotrophic methanogenesis in archaeal phylum Verstraetearchaeota
reveals the shared ancestry of all methanogens. *Proc. Natl. Acad. Sci. USA* **116**, 5037-5044 (2019).

Boyd, J. A., et al. Methyl-coenzyme M reductase genes of a deep-subseafloor *Archaeoglobi*. *ISME J.*
**13**, 1269-1279 (2019).

Chen, S., et al. Anaerobic oxidation of ethane by archaea from a marine hydrocarbon seep. *Nature* **568**,
108–111 (2019).

Csűrös, M., & Miklós, I. Streamlining and large ancestral genomes in Archaea inferred with a
phylogenetic birth-and-death model. *Mol. Biol. Evol.* **26**, 2087-2095 (2009).

Csűös, M. Count: evolutionary analysis of phylogenetic profiles with parsimony and likelihood.
*Bioinformatics* **26**, 1910–1912 (2010).

Evans, P. N., et al. An evolving view of methane metabolism. *Nature Rev Microbiol.* In Press (2019).

Hua, Z. S., et al. Genomic inference of the metabolism and evolution of the archaeal phylum
Aigarchaeota. *Nat. Commun.* **9**, 2832 (2018).

Kalyaanamoorthy, S., Minh, B. Q., Wong, T. K., von Haeseler, A., & Jermin, L. S. ModelFinder: fast
model selection for accurate phylogenetic estimates. *Nat. Methods* **14**, 587 (2017).

Mall, A., et al. Reversibility of citrate synthase allows autotrophic growth of a thermophilic

bacterium. *Science* **359**, 563-567 (2018).

Nunoura, T., et al. A primordial and reversible TCA cycle in a facultatively chemolithoautotrophic
thermophile. *Science* **359**, 559-563 (2018).

Laso-Pérez, R., et al. Thermophilic archaea activate butane via alkyl-coenzyme M formation. *Nature*
**539**, 396-401 (2016).

Pang, T. Y., & Lercher, M. J. Each of 3,323 metabolic innovations in the evolution of *E. coli* arose
through the horizontal transfer of a single DNA segment. *Proc. Natl. Acad. Sci. USA* **116**, 187-192
(2019).

Parks, D., Imelfort, M., Skennerton, C., Hugenholtz, P., Tyson, G. (2015). CheckM: assessing the
quality of microbial genomes recovered from isolates, single cells, and metagenomes. *Genome*
*Res.* **7**, 1043-1055 (2015).

Parks, D., et al. A standardized bacterial taxonomy based on genome phylogeny substantially revises
the tree of life. *Nat. Biotechnol.* **36**, 996-1004 (2019).

Ronquist, F., et al. MrBayes 3.2: efficient Bayesian phylogenetic inference and model choice across a
large model space. *Syst. Biol.* **61**, 539-542 (2012).

Schulz, F., et al. Giant viruses with an expanded complement of translation system
components. *Science* **356**, 82-85 (2017).

Sorokin, D. Y., et al. Discovery of extremely halophilic, methyl-reducing euryarchaea provides insights
into the evolutionary origin of methanogenesis. *Nat. Microbiol.* **2**, 17081 (2017).

Wolf, Y. I., Makarova, K. S., Yutin, N., & Koonin, E. V. Updated clusters of orthologous genes for
Archaea: a complex ancestor of the Archaea and the byways of horizontal gene transfer. *Biol.*
*Direct* **7**, 46 (2012).

Reviewers' comments:

Reviewer #1 (Remarks to the Author):

The authors addressed the main concerns from the reviews. The revised version of the manuscript appears to have improved, in particular regarding the phylogenomic reconstructions.

Typo corrections:

line 137, 209 and other places: Wood-Ljungdhal - replace with Wood-Ljungdahl

line 218: neofunctionalisation - replace with neofunctionalisation

Reviewer #2 (Remarks to the Author):

Many thanks to the authors for the implemented changes and comments. I especially appreciate that the authors provide a tree file for the species tree, as it does make cross-referencing between individual parts of the manuscript much easier.

1. Regarding the author's responses about rooting the trees: The authors incorrectly cite Williams et al., 2017 when they state that trees were rooted based on the assumption that TACK are evolutionary more ancient. Williams places "the root between DPANN and a clade comprising the Euryarchaeota and TACK/Lokiarchaeum lineages". Based on the tree that is shown in Fig. 3A in Williams et al., Euryarchaeota would be considered more ancient than TACK. Other papers cited in this article place the root between Euryarchaeota and TACK or between most of the Euryarchaeota and TACK plus some Thermococcales/some methanogens (Refs 9,10,18 in Williams et al., 2017). Additionally, stating that the root was chosen because TACK contain the greatest taxonomic diversity (Line 284) is also not a good way of determining the root since (a) microbial lineages can diversify at later points and (b) looking at the overall branch length shown in the species tree (Fig. 1) TACK does not necessarily appear to be more diverse than the Euryarchaeota. Therefore, there seems to be not much support for TACK being more ancient and I do not think the authors can make an argument for choosing this position of the root. Since the archaeal root is still debated right now, the authors could consider an alternative method such as outgroup-free rooting methods (i.e. Williams et al., 2015 or Coleman et al., 2018).

2. Regarding previous Comment 2 that discusses the vertical versus horizontal decent of the *mcr* gene. I still would argue that especially for the TACK the history of *mcr* might be more complicated than discussed in the manuscript. More specifically, there seem to be more examples other than Archaeoglobus and the Bathyarchaeota (Line 298) with a more complicated history and potential HGT events and that include Korarchaeota or the recently published Helarchaeota within Asgard (Seitz et al., 2019). Another example not discussed are the Arc I group archaea that group close to ANME-1 in the gene but not the species tree. Maybe showing a collapsed species tree next to the *mcr* gene tree might help to better visualize vertical decent versus HGT?

3. This is linked to the previous comment: It appears that not all species that should encode for the *mcr* gene and that are present in the species tree are also present in the *mcr* gene tree. For example, the type strain Methanopyrus kandleri has a *mcr* gene (Nölling et al., 1996) and should be also present in the Methanopyri genomes included in the species tree (i.e. for GCF_002201915.1 the gene WP_088335801.1 seems to correspond to *mcrA*). This specific enzyme turns up in NCBI as coenzyme-B_sulfoethylthiotransferase alpha subunit, which is maybe why it was not picked up, but it should be included to make sure that the species and gene tree can be compared to each other as exactly as

possible and it should be cross-checked whether only Methanopyri or also other taxa are not included.

4. Response to comment 8: I am aware of experimental evidence of the citrate synthase acting in reverse. However, it is important to acknowledge that Mall et al., 2018 also state that ‘the roTCA cycle can hardly be recognized bioinformatically’ and to my knowledge the reversibility of this gene has not been shown in archaea so far. Since there is no good way to distinguish between the roTCA and TCA, I would not favor one over the other and not only mention the roTCA in Figure 2. Similarly, the statement in Lines 236-238 would more argue for the presence of the TCA cycle, while not being able to completely rule out that this pathway also might act in reverse as shown for bacteria.

5. Line 54 ‘these genes’ can be ambiguous, could the authors therefore consider exchanging this with the gene names.

6. Lines 129-132: With the attached and updated tree file and Figure 1 this becomes a bit clearer to see, therefore, I added this new comment.

a) What MAGs do the authors specifically refer to with ‘the placement of these MAGs also reveals deep branching within the respective lineages’. Looking at the treefile the new Hadesarchaeota and Archaeoglobales MAGs branch within other Hadesarchaeota and Archaeoglobales. Also in the zoomed in version for the Verstraetearchaeota, Methansuratus, which belongs to the Ca. Verstraetearchaeota, seems to branch deeper than the new bins. Therefore, not all new lineages appear to be deep-branching. Additionally, see my previous comments about Euryarchaeota being more “derived”. Altogether, I would suggest some clearer wording.

b) The Hadesarchaea MAGs bin_199 and bin_103 seem to fall inside the Hadesarchaeota (at least in the species tree). Therefore, another question is, whether the rooting for Figure S7a is accurate.

7. Typo Fig S3: Verstraetearchaeota

8. SI Material Line 143: ‘How water Crenchaetotic Group III’ should be ‘hot water crenarchaeotic group’

9. New Table S2/Main text Lines 276-279: The authors performed a Mann-Whitney test to test whether the mcr contig has a different GC content than the rest of the genome. However, since the sample size (n) seems to be one genome it is not appropriate to perform such a test for this kind of data (even when combining the genomes from the different phyla/lineages the sample size remains too small). On average the mcr seems to have a difference of 3% to the rest of the genome (in one case even 7%). This by far exceeds the natural variation within the genome looking at the SE values and thus the mcr “looks” rather different from the rest of the genome. Therefore, I would suggest rewriting this section (and removing the statistics).

References

1. Coleman, G. A., Pancost, R. D. & Williams, T. A. Resolving the origins of membrane phospholipid biosynthesis genes using outgroupfree rooting. *bioRxiv* 312991 (2018). doi:10.1101/312991
2. Nölling, J. et al. NOTES: Phylogeny of *Methanopyrus kandleri* Based on Methyl Coenzyme M Reductase Operons. *International Journal of Systematic and Evolutionary Microbiology* 46, 1170–1173 (1996).
3. Williams, T. A. et al. New substitution models for rooting phylogenetic trees. *Phil. Trans. R. Soc. B* 370, 20140336 (2015).

Reviewer #3 (Remarks to the Author):

Title: Insights into the ecological roles and evolution of mcr-containing hot spring Archaea

The authors have responded sincerely to my previous comments, The manuscript in general and the figures are very clear and informative. There are just one or two points which they may have overlooked. They are trivial, and my explanation may have been too vague.

6) Other points

The arrow from CO in the carbon monoxide dehydrogenase reaction should curve right, not left.

The arrow is still curving left. (I mean the vertical arrow below CO) The methyl group and the CO combine (with CoA) to produce acetyl-CoA. In the present figure, it looks like CO is released from the methyl group.

In addition, change Glyconeogenesis to Gluconeogenesis.

I have no further comments.

Responses to the reviewer's comments

Reviewer #1 (Remarks to the Author):

The authors addressed the main concerns from the reviews. The revised version of the manuscript appears to have improved, in particular regarding the phylogenomic reconstructions.

Typo corrections:

Comment 1: line 137, 209 and other places: Wood-Ljungdhal - replace with Wood-Ljungdahl

Response: Revised as suggested.

Comment 2: line 218: neofunctionalisation - replace with neofunctionalisation

Response: Revised as suggested.

Reviewer #2 (Remarks to the Author):

Many thanks to the authors for the implemented changes and comments. I especially appreciate that the authors provide a tree file for the species tree, as it does make cross-referencing between individual parts of the manuscript much easier.

Comment 1. Regarding the author's responses about rooting the trees: The authors incorrectly cite Williams et al., 2017 when they state that trees were rooted based on the assumption that TACK are evolutionary more ancient. Williams places "the root between DPANN and a clade comprising the *Euryarchaeota* and TACK/*Lokiarchaeum* lineages". Based on the tree that is shown in Fig. 3A in Williams et al., *Euryarchaeota* would be considered more ancient than TACK. Other papers cited in this article place the root between *Euryarchaeota* and TACK or between most of the *Euryarchaeota* and TACK plus some *Thermococcales*/some methanogens (Refs 9,10,18 in Williams et al., 2017). Additionally, stating that the root was chosen because TACK contain the greatest taxonomic diversity (Line 284) is also not a good way of determining the root since (a) microbial lineages can diversify at later points and (b) looking at the overall branch length shown in the species tree (Fig. 1) TACK does not necessarily appear to be more diverse than the *Euryarchaeota*. Therefore, there seems to be not much support for TACK being more ancient and I do not think the authors can make an argument for choosing this position of the root. Since the archaeal root is still debated right now, the authors could consider an alternative method such as outgroup-free rooting methods (i.e. Williams et al., 2015 or Coleman et al., 2018).

Response: We thank the reviewer for the suggestion and we agree with the point related to the outgroup-free rooting methods. In the revised Manuscript, we have adopted two complementary approaches to root the concatenated and individual *mcrABG* gene trees including BEAST and minimal ancestor deviation (MAD) rooting method which was used in the suggested paper (Coleman et al., 2018). This analysis suggests the possibility that methanogens might be diversified from alkanotrophs. However, we can't rule out the possibility of long-branching attraction effect which resulting in a problematic placement of root. Also, the bootstrap confidence of 53% at the root of the two main lineages is low, indicating the weak evidence of proper placement of root. Even so, the topology for methane metabolizing archaea is still the same as the previous version. Therefore, it result supports our

inference of HGT being rare among methane oxidizers and methanogens, but a ‘frequent’ occurrence in alkanotrophs.

Comment 2. Regarding previous Comment 2 that discusses the vertical versus horizontal decent of the *mcr* gene. I still would argue that especially for the TACK the history of *mcr* might be more complicated than discussed in the manuscript. More specifically, there seem to be more examples other than *Archaeoglobus* and the *Bathyarchaeota* (Line 298) with a more complicated history and potential HGT events and that include *Korarchaeota* or the recently published *Helarchaeota* within Asgard (Seitz et al., 2019). Another example not discussed are the Arc I group archaea that group close to ANME-1 in the gene but not the species tree. Maybe showing a collapsed species tree next to the *mcr* gene tree might help to better visualize vertical decent versus HGT?

Response: As suggested, we have added the collapsed species tree (Fig. 3b) to the Fig 3 to facilitate the visualization of HGT and vertical decent events. Both trees show high similarity of topology for the TACK lineages except in MAGs from the *Bathyarchaeota* are distant from euryarchaeotal alkane oxidizers. HGT seems to be an important source of gene innovation in alkanotrophs. Therefore when compared to methane metabolizing microbes, we observed more ‘frequent’ HGT events in alkanotrophs. However, we do still think that HGTs are rare among methane metabolizing and TACK lineages. For example, the paper where *mcr* genes are reported in the *Korarchaeota* support our conclusion since they mentioned methane metabolism might be originated in this phylum (McKay et al., 2019). We have added in the ANME-1 and Arc I lineage Mcr/species tree discrepancy for an example of methanogen/methanotroph. For completeness we have also added in Gom-ArcI and ANME-2d we have added in a sentence to identify HGT between an alkane oxidizing *Euryarchaeota* and Gom-ArcI.

Comment 3. This is linked to the previous comment: It appears that not all species that should encode for the *mcr* gene and that are present in the species tree are also present in the *mcr* gene tree. For example, the type strain *Methanopyrus kandleri* has a *mcr* gene (Nölling et al., 1996) and should be also present in the *Methanopyri* genomes included in the species tree (i.e. for GCF_002201915.1 the gene WP_088335801.1 seems to correspond to *mcrA*). This specific enzyme turns up in NCBI as coenzyme-B_sulfoethylthiotransferase alpha subunit, which is maybe why it was not picked up, but it should be included to make sure that the species and gene tree can be compared to each other as exactly as possible and it should be cross-checked whether only *Methanopyri* or also other taxa are not included.

Response: This has been changed as suggested. In the revised version of Fig. 1 and 3, *Methanopyrus kandleri*, *Helarchaeota* Hel_GB_A, *Helarchaeota* Hel_GB_B, and *Ca. Agroarchaeum ethanivorans* have been added. We observed that the placement of added Mcr sequences of *Methanopyri* in gene tree show similar positions as in the species tree which puts them near *Methanococcales* and *Methanobacteriales* lineages. Among the highly diversified alkanotrophs, the McrABG sequences from in ASGARD locate between *Ca. Syntrophoarchaeum* and *Bathyarchaeota* Mcr sequences. Based on the parsimony inference, we speculate that ‘frequent’ HGT events have occurred from euryarchaeal to *Bathyarchaeota* and Asgard members, with potential HGT from euryarchaeal donor to *Hadesarchaeota* organisms.

Comment 4. Response to comment 8: I am aware of experimental evidence of the citrate synthase acting in reverse. However, it is important to acknowledge that Mall et al., 2018 also state that ‘the roTCA cycle can hardly be recognized bioinformatically’ and to my knowledge the reversibility of this gene has not been shown in archaea so far. Since there is no good way to distinguish between the roTCA and TCA, I would not favor one over the other and not only mention the roTCA in Figure 2. Similarly, the statement in Lines 236-238 would more argue for the presence of the TCA cycle, while not being able to completely rule out that this pathway also might act in reverse as shown for bacteria.

Response: Revised as suggested. We removed all the descriptions related to roTCA cycle, including the sentence in the Main text and Figure 2.

Comment 5. Line 54 ‘these genes’ can be ambiguous, could the authors therefore consider exchanging this with the gene names.

Response: Here, “these genes” means *mcrABG* genes, which has been revised in the revised Manuscript.

Comment 6. Lines 129-132: With the attached and updated tree file and Figure 1 this becomes a bit clearer to see, therefore, I added this new comment.

a) What MAGs do the authors specifically refer to with ‘the placement of these MAGs also reveals deep branching within the respective lineages’. Looking at the treefile the new *Hadesarchaeota* and *Archaeoglobales* MAGs branch within other *Hadesarchaeota* and *Archaeoglobales*. Also in the zoomed in version for the *Verstraetearchaeota*, *Methansuratus*, which belongs to the *Ca. Verstraetearchaeota*, seems to branch deeper than the new bins. Therefore, not all new lineages appear to be deep-branching. Additionally, see my previous comments about *Euryarchaeota* being more “derived”. Altogether, I would suggest some clearer wording.

Response: We agree with reviewer that not all MAGs in this study are the deepest branches in the corresponding lineages. However, after a lot of recent reported MAGs have been added and analyzed in this study many of these lineages are basal or near basal. But here we aim to emphasize that methane /alkane metabolism ability may have evolved earlier than we thought previously since many lineages in TACK and *Euryarchaeota* phyla now appear contain at least one member with *mcr* genes. Even though some of our MAGs are not the deepest branch, other *mcr*-containing MAGs from other studies could support this. For instance, *Methanosuratus*, as mentioned by reviewer, was detected to contain *mcrABG* genes and located deeper than our MAGs. But as all current members from this lineage have *mcrABG* genes which suggest that although our genome is not basal, a MAG from another study with *mcrABG* is still basal. Also, *mcrABG* containing MAGs from the *Archaeoglobi* seems fall inside other genome clades in this class, however, *Ca. Polytropus marifundus* locates outermost to all these lineages (Fig. S4), again suggesting a *mcrABG* containing MAG is basal. This same situation also occurred in *Hadesarchaeota* which has been explained below. We have reworded these sentences to avoid the ambiguity.

b) The *Hadesarchaeota* MAGs bin_199 and bin_103 seem to fall inside the *Hadesarchaeota* (at least in the species tree). Therefore, another question is, whether the rooting for Figure S7a is accurate.

Response: The observed discrepancy between Fig 1 and Fig S7a is mainly caused by the different references selected for the tree construction. Several newly published genomes (WYZ-4, -5 and -6) were added to Fig. 1; but not to Fig S7a since the first manuscript submission. To avoid the confusion, we have revised Fig S7a, to include newly published MAGs from the *Hadesarchaeota* (and match those from Fig. 1). Also, we agree that our *mcr*-containing MAGs fall within the *Hadesarchaeota*, but even though our MAGs are not the deepest, other *mcr*-containing MAGs placed at basal positions of these lineages (this comment is similar to Comment 6a).

Comment 7. Typo Fig S3: *Verstraetearchaeota*

Response: Revised as suggested.

Comment 8. SI Material Line 143: ‘How water Crenchaotic Group III’ should be ‘hot water crenarchaeotic group’

Response: Revised as suggested.

Comment 9. New Table S2/Main text Lines 276-279: The authors performed a Mann-Whitney test to test whether the *mcr* contig has a different GC content than the rest of the genome. However, since the sample size (n) seems to be one genome it is not appropriate to perform such a test for this kind of data (even when combining the genomes from the different phyla/lineages the sample size remains too small). On average the *mcr* seems to have a difference of 3% to the rest of the genome (in one case even 7%). This by far exceeds the natural variation within the genome looking at the SE values and thus the *mcr* “looks” rather different from the rest of the genome. Therefore, I would suggest rewriting this section (and removing the statistics).

Response: We agree with the reviewer and have rewritten this section. For this test we used McrABG genes to compare the GC content of whole genome in the previous version of manuscript. We also think this might be the main reason that leads to the higher variation since the total length of the three genes are pretty short (< 4Kbp). If we use the *mcr*-containing contigs to compare to the rest of the genomes, their GC content are really close which as listed in the revised Supplementary Table 2. The highest variation was observed in *Hadesarchaeota*, which support our hypothesis that HGT frequently occurred among alkanotrophs but rarely detected in methane metabolizing microbes.

References

1. Coleman, G. A., Pancost, R. D. & Williams, T. A. Resolving the origins of membrane phospholipid biosynthesis genes using outgroupfree rooting. *bioRxiv* 312991 (2018). doi:10.1101/312991
2. Nölling, J. et al. NOTES: Phylogeny of *Methanopyrus kandleri* Based on Methyl Coenzyme M Reductase Operons. *International Journal of Systematic and Evolutionary Microbiology* 46, 1170–1173 (1996).
3. Williams, T. A. et al. New substitution models for rooting phylogenetic trees. *Phil. Trans. R. Soc. B* 370, 20140336 (2015).

Reviewer #3 (Remarks to the Author):

Title: Insights into the ecological roles and evolution of mcr-containing hot spring Archaea

The authors have responded sincerely to my previous comments, The manuscript in general and the figures are very clear and informative. There are just one or two points which they may have overlooked. They are trivial, and my explanation may have been too vague.

Comment 1: 6) Other points

The arrow from CO in the carbon monoxide dehydrogenase reaction should curve right, not left.

The arrow is still curving left. (I mean the vertical arrow below CO) The methyl group and the CO combine (with CoA) to produce acetyl-CoA. In the present figure, it looks like CO is released from the methyl group.

Response: We apologize for the misunderstanding. Revised as suggested.

Comment 2: In addition, change Glyconeogenesis to Gluconeogenesis.

Response: Revised as suggested.

Reviewer #2 (Remarks to the Author):

Many thanks to the authors for the comments and added changes. I just have some very minor comments that are outlined below.

Minor comments main manuscript:

1. Line 51: Typo, consider '... suggests a hydrothermal origin for the microorganisms based on optimal growth temperature'.
2. Line 56: '...methanogenesis at high temperature likely existed ...'
3. Line 89: '...using bioinformatic methods'
4. Line 123 (and related sections throughout the manuscript and figures): The two public genomes that branch "sister to the Crenarchaeota" have been named in a very recent publication that is also cited by the authors as Nezharchaeota (based on the MAGs WYZ-LMO8 and WYZ-LMO7; Wang et al., 2019). For consistency I would suggest to use the same name for that lineage throughout the manuscript.
5. Line 139: 'based on the absence'
6. Regarding previous comment one and now Lines 499-503: First of all, many thanks to the authors for implementing these methods. The only thing that could be done here is to add a short sentence were those two methods place the root and whether they are consistent.

Minor comments Figures

7. Caption Figure S8: 'see methods for detailed settings' and 'all bootstrap values'

Responses to the reviewer's comments

Reviewer #2 (Remarks to the Author):

Many thanks to the authors for the comments and added changes. I just have some very minor comments that are outlined below.

Minor comments main manuscript:

Comment 1: Line 51: Typo, consider '... suggests a hydrothermal origin for the microorganisms based on optimal growth temperature'.

Response: Revised as suggested.

Comment 2: Line 56: '...methanogenesis at high temperature likely existed ...'

Response: Revised as suggested.

Comment 3: Line 89: '...using bioinformatic methods'

Response: Revised as suggested.

Comment 4: Line 123 (and related sections throughout the manuscript and figures): The two public genomes that branch "sister to the Crenarchaeota" have been named in a very recent publication that is also cited by the authors as Nezharchaeota (based on the MAGs WYZ-LMO8 and WYZ-LMO7; Wang et al., 2019). For consistency I would suggest to use the same name for that lineage throughout the manuscript.

Response: Revised as suggested.

Comment 5: Line 139: 'based on the absence'

Response: Revised as suggested.

Comment 6: Regarding previous comment one and now Lines 499-503: First of all, many thanks to the authors for implementing these methods. The only thing that could be done here is to add a short sentence were those two methods place the root and whether they are consistent.

Response: Revised as suggested.

Minor comments Figures

Comment 7: Caption Figure S8: 'see methods for detailed settings' and 'all bootstrap values'

Response: Revised as suggested.